# GD$^2$: Robust Graph Learning under Label Noise via Dual-View Prediction Discrepancy

**Kailai Li**[1†], **Jiong Lou**[1†], **Jiawei Sun**[1], **Honghong Zeng**[1], **Wen Li**[5,*], **Chentao Wu**[1,3,4],
**Yuan Luo**[1], **Wei Zhao**[2], **Shouguo Du**[6], **Jie Li**[1,3,4,*]

[1]Department of Computer Science and Engineering, Shanghai Jiao Tong University
[2]Shenzhen University of Advanced Technology
[3]Yancheng Blockchain Research Institute
[4]Shanghai Key Laboratory of Trusted Data Circulation and Governance and Web3
[5]Shanghai University of International Business and Economics
[6]Shanghai Municipal Big Data Center
{kailai_li, lj1994, noelsjw, zhhhhh5}@sjtu.edu.cn,
wen.li.sh@outlook.com, {wuct,luoyuan}@cs.sjtu.edu.cn,
{weizhao86, shouguo.du.sh}@outlook.com, lijiecs@sjtu.edu.cn

## Abstract

Graph Neural Networks (GNNs) achieve strong performance in node classification tasks but exhibit substantial performance degradation under label noise. Despite recent advances in noise-robust learning, a principled approach that exploits the node-neighbor interdependencies inherent in graph data for label noise detection remains underexplored. To address this gap, we propose GD$^2$, a noise-aware Graph learning framework that detects label noise by leveraging Dual-view pre-diction Discrepancies. The framework contrasts the *ego-view*, constructed from node-specific features, with the *structure-view*, derived through the aggregation of neighboring representations. The resulting discrepancy captures disruptions in semantic coherence between individual node representations and the structural context, enabling effective identification of mislabeled nodes. Building upon this insight, we further introduce a view-specific training strategy that enhances noise detection by amplifying prediction divergence through differentiated view-specific supervision. Extensive experiments on multiple datasets and noise settings demonstrate that GD$^2$ achieves superior performance over state-of-the-art baselines.

## 1 Introduction

Graph Neural Networks (GNNs) have recently achieved significant advancements in node classification, with extensive applicability across domains such as social network analysis [9], recommender systems [38], and biomedical research [36]. Nevertheless, obtaining high-quality labeled data remains challenging in real-world scenarios, and noisy labels are inevitably present. Prior research indicates that noisy labels can mislead the training of GNNs, substantially degrading predictive accuracy and generalization capability [45]. Furthermore, due to the message-passing mechanism inherent in GNNs, the negative impact of label noise is exacerbated [5, 28]. Consequently, developing robust GNN models under label noise represents a crucial and pressing research objective.

To alleviate negative impacts associated with noisy labels, general noise-learning approaches commonly exploit the memorization effect exhibited by neural networks [1, 42] or rely on handcrafted

---

*Corresponding authors: Wen Li, Jie Li.
†Equal contribution.

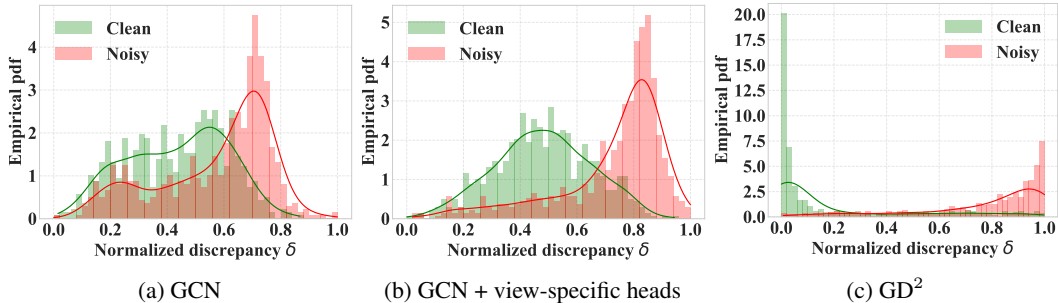

Figure 1: Normalized prediction discrepancy between the ego-view and structure-view on the Computer dataset. (a) Standard GCN yields observable but limited separation in the discrepancy distribution between clean and noisy nodes, hindering effective noise detection. (b) GCN with a shared encoder and view-specific classifier heads produces a more distinguishable discrepancy distribution. (c) $GD^2$ enhances the distributional separability in prediction discrepancies between clean and noisy nodes, yielding a well-separated distribution and facilitating noise detection.

heuristic signals [27, 10, 14] for filtering or rectifying mislabeled samples. However, these methods implicitly assume that the data are independent and identically distributed (i.i.d.), which does not hold in graph data due to interdependencies among connected nodes. Consequently, existing noise-learning techniques perform suboptimally when directly applied to node classification tasks [32]. Recently, several methods specifically tailored for graph data have emerged, incorporating structural informa-tion into noise-robust learning. Representative methods include graph structure refinement [5, 6], structure-based curriculum learning [39, 35], label correction via propagation [4, 2], and incorporation of auxiliary supervision [28, 8, 43]. A common trait among these approaches is the implicit use of semantic alignment between node features and the structural context. Such alignment captures the node–neighbor interdependencies inherent to graph data, contributing to robust representation learning and label noise mitigation. However, most existing techniques inherit noise detection mecha-nisms originally developed for i.i.d. data and do not explicitly leverage such relational patterns. The potential of interdependencies for identifying noisy labels remains underexplored. Consequently, an open question naturally arises: *can the intrinsic dependencies between node features and structural context be explicitly modeled to enable effective label noise detection in graph data?*

To address the aforementioned research question, this paper presents $GD^2$, a noise-aware graph learning framework that explicitly leverages the semantic discrepancies between node features and structural context to detect label noise. The core component is a noise detection module based on dual-view prediction discrepancy. Specifically, node representations are decoupled into two distinct views based on information source. The *ego-view* is constructed solely from node-specific features and captures individual semantic information. The *structure-view* is derived by aggregating neighbor representations and encodes contextual semantics from local graph topology. The prediction discrepancy between the two views is exploited as an intrinsic signal for identifying potential label errors. Figure 1 illustrates the prediction discrepancy between the ego-view and structure-view under label noise. The key insight is that label noise disrupts semantic coherence between the two views, leading to divergent predictions. The ego-view, relying solely on individual features, tends to overfit spurious correlations introduced by incorrect labels, resulting in semantically misaligned representations. In contrast, the structure-view captures topological patterns that remain relatively stable under noise. Therefore, prediction divergence between the two views serves as an indicator for identifying mislabeled nodes. This insight is further supported by theoretical analysis, which reveals that clean-labeled nodes inherently exhibit bounded cross-view representation discrepancies. Building upon the above noise detection mechanism, we also introduce a view-specific training strategy, which provides differentiated supervision to each view to further enhance prediction discrepancies and facilitate noise detection. The main contributions are summarized as follows:

**(1)** We propose a novel perspective for noisy label detection in graph data, that prediction discrepancy between ego-view and structure-view can be an indicator for identifying label errors.

**(2)** We introduce a graph learning framework named $GD^2$, which incorporates a noise detection module and a view-specific training strategy. By quantifying predictive discrepancies between views, the proposed framework effectively detects mislabeled nodes and facilitates robust learning.

**(3)** Extensive experiments on multiple real-world graph datasets under varying noise levels demonstrate that GD$^2$ consistently outperforms state-of-the-art baselines, highlighting its effectiveness in addressing label noise in graph learning.

## 2 Related Work

### 2.1 Noisy Label Detection

Noisy label detection aims to identify and remove incorrectly labeled samples from the training data. The main challenge lies in defining a suitable metric to quantify the label quality. A widely-adopted approach leverages the memorization effect of deep neural networks [1], utilizing criteria such as training loss or prediction confidence. Representative methods include Decoupling [23], Co-teaching [13], Co-teaching+ [42], and ProMix [41]. More advanced criteria have also been developed, such as AUM [27], which identifies mislabeled samples by tracking the prediction margin during training, TopoFilter [37], which isolates noisy samples based on topological properties in the representation space, and OT-Filter [10], which selects samples from the perspective of optimal transport theory.

Recently, some studies have extended sample selection methods to graph-structured data. For instance, TSS [39] employs class-conditional betweenness centrality based on graph topology to select nodes; UnionNet [19] aggregates labels to perform sample re-weighting and label correction; and CLN-ode [35] introduces a multi-perspective difficulty measurer for detecting label noise. Nevertheless, these methods typically inherit noise detection mechanisms originally developed for i.i.d. data, failing to explicitly model intrinsic dependencies between node features and structural context for effective noisy label detection.

### 2.2 Noise Robust Learning

Besides detecting noisy labels, another line of research focuses on developing robust training algorithms and models. Such methods include designing noise-resistant network architectures [17, 11], robust loss functions [46], and regularization [34, 24], as well as training strategies [15, 20, 40].

In the context of Graph Neural Networks (GNNs), previous studies [5, 8] have demonstrated that label noise can propagate through the graph structure, significantly impairing the performance of GNNs. Several methods have mitigated this issue by refining graph structure. For instance, NRGNN [5] connects unlabeled nodes to trustworthy labeled neighbors to alleviate the effects of label noise, while RS-GNN [6] learns a denoised graph structure to reduce noise propagation. Other methods introduce auxiliary supervisory signals to strengthen robustness, including self-training and consistency regularization [28], information-theoretic objectives [49], and pairwise consistency modeling [8]. Graph contrastive learning has also been leveraged, with representative methods including RNCGLN [50], ALEX [43], and CGNN [44]. Label propagation approaches have likewise been explored: ERASE [2] integrates label propagation with coding rate minimization, while R$^2$LP [4] applies propagation on reconstructed graphs to correct noisy labels. DND-NET [7] mitigates noise by decoupling message passing in GNNs.

Although these methods have improved the robustness of GNNs against noisy labels, their identification of mislabeled samples still predominantly relies on loss-based or confidence-based metrics. Different from existing approaches, we propose a graph learning framework that identifies noisy labels based on dual-view prediction discrepancy.

## 3 Preliminary

This work focuses on node classification under label noise. Consider a graph $\mathcal{G} = (\mathcal{V}, \mathcal{E})$, where $\mathcal{V}$ denotes the set of nodes and $\mathcal{E} \subseteq \mathcal{V} \times \mathcal{V}$ represents the set of edges. The graph $\mathcal{G}$ is assumed to be undirected and without self-loops. Node features are represented by a matrix $\mathbf{X} \in \mathbb{R}^{|\mathcal{V}| \times d}$, where $|\mathcal{V}|$ is the number of nodes and $d$ is the feature dimension. Let $\mathbf{Y} \in \mathbb{R}^{|\mathcal{V}| \times C}$ denote the one-hot encoded ground-truth label matrix, and $\tilde{\mathbf{Y}} \in \mathbb{R}^{|\mathcal{V}| \times C}$ the one-hot encoded noisy label matrix, where $C$ is the number of classes. The adjacency matrix is denoted by $\mathbf{A} \in \mathbb{R}^{|\mathcal{V}| \times |\mathcal{V}|}$, where $\mathbf{A}_{ij} = 1$ if nodes $v_i$ and $v_j$ are connected, and $\mathbf{A}_{ij} = 0$ otherwise. The objective is to train a Graph Neural Network

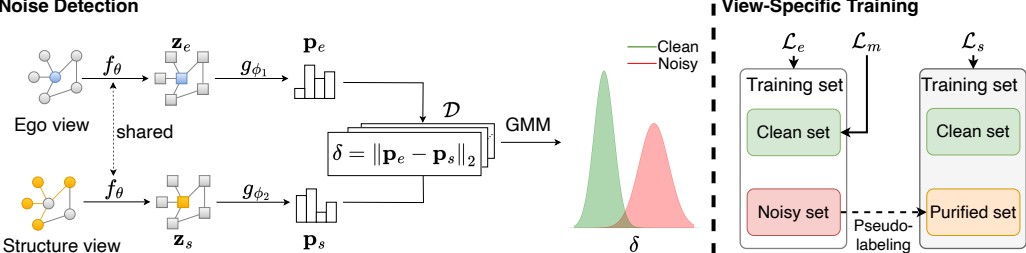

**Figure 2:** Overview of the proposed framework GD$^2$, which consists of two core modules: noise detection and view-specific training. In the noise detection module, the cross-view prediction discrepancy is quantified and modeled using a Gaussian Mixture Model (GMM) to identify label noise. For model training, pseudo-labels are assigned to confident noisy nodes to refine supervision. The framework then jointly optimizes three view-specific objectives. The two modules are mutually reinforcing: the denoised training set improves model robustness, which in turn sharpens the discrepancy for more effective noise detection.

(GNN) on the training set $\mathcal{V}_{\text{train}}$ with noisy labels $\tilde{\mathbf{Y}}$, such that accurate predictions can be made for the true labels $\mathbf{Y}$ on the unlabeled nodes.

Graph Neural Networks (GNNs) typically follow a message-passing architecture, where node representations are updated by aggregating information from neighboring nodes. At the $\ell$-th layer, the representation of node $v$ is computed as:

$$\mathbf{h}_v^{(\ell)} = \text{UPDATE}^{(\ell)}\left(\mathbf{h}_v^{(\ell-1)}, \text{AGG}^{(\ell)}\left(\{\mathbf{h}_u^{(\ell-1)} \mid u \in \mathcal{N}(v)\}\right)\right), \tag{1}$$

where $\mathcal{N}(v)$ denotes the set of neighbors of node $v$, $\text{AGG}^{(\ell)}$ is the aggregation function, and $\text{UPDATE}^{(\ell)}$ is the update function. Initial representations are given by node features, i.e., $\mathbf{h}_v^{(0)} = \mathbf{x}_v \in \mathbb{R}^d$. For Graph Convolutional Networks [16], the representation matrix at the $\ell$-th layer is updated as:

$$\mathbf{H}^{(\ell)} = \sigma\left(\tilde{\mathbf{D}}^{-\frac{1}{2}}\tilde{\mathbf{A}}\tilde{\mathbf{D}}^{-\frac{1}{2}}\mathbf{H}^{(\ell-1)}\mathbf{W}^{(\ell)}\right), \tag{2}$$

where $\tilde{\mathbf{A}} = \mathbf{A} + \mathbf{I}$ is the adjacency matrix with self-loops, $\mathbf{I}$ is the identity matrix, $\tilde{\mathbf{D}}$ is the degree matrix of $\tilde{\mathbf{A}}$, $\mathbf{W}^{(\ell)}$ is the learnable weight matrix, and $\sigma$ denotes a non-linear activation function.

## 4 Method

This section details the proposed framework GD$^2$. A standard approach to learning under label noise typically involves two stages: first identifying mislabeled samples, and then training the model using the samples estimated to be clean. As illustrated in Figure 2, GD$^2$ follows this paradigm and consists of two core components: a noise detection module and a view-specific training module. In the noise detection module, cross-view prediction discrepancies are used to estimate the likelihood of label correctness via a Gaussian Mixture Model. In the training module, pseudo-labels are generated from a mixed-view, and then distinct training objectives are designed for different views. The two components are trained in a mutually reinforcing manner: the detection module provides a cleaner training set for the model, while the training process strengthens view discrepancies, further improving noise detection. The pseudo code of GD$^2$ is presented in the Appendix E.

### 4.1 Noise Detection via Prediction Discrepancy

Identification of mislabeled nodes is achieved by explicitly modeling prediction discrepancies between different views. The distinct views differ in information sources: the ego-view relies exclusively on intrinsic node features, whereas the structure-view aggregates neighborhood information. Node representations of the ego-view and structure-view, denoted as $\mathbf{Z}_e$ and $\mathbf{Z}_s$, respectively, are first computed. Specifically, a Graph Convolutional Network (GCN)-based encoder $f_\theta$ with parameters $\theta$ is employed. Representations at the $\ell$-th layer for the ego-view and structure-view, denoted by $\mathbf{H}_e^{(\ell)}$

and $\mathbf{H}_s^{(\ell)}$, are updated according to:

$$\mathbf{H}_e^{(\ell)} = \sigma\left(\mathbf{H}_e^{(\ell-1)}\mathbf{W}^{(\ell)}\right), \quad \mathbf{H}_s^{(\ell)} = \sigma\left(\mathbf{D}^{-\frac{1}{2}}\mathbf{A}\mathbf{D}^{-\frac{1}{2}}\mathbf{H}_s^{(\ell-1)}\mathbf{W}^{(\ell)}\right), \tag{3}$$

where $\mathbf{W}^{(\ell)}$ denotes learnable weights at layer $\ell$, $\sigma$ represents a non-linear activation function, $\mathbf{A}$ is the adjacency matrix without self-loops, and $\mathbf{D}$ is the corresponding degree matrix of $\mathbf{A}$. The final representations for both views are:

$$\mathbf{Z}_e = f_\theta(\mathbf{X}, \mathbf{I}), \quad \mathbf{Z}_s = f_\theta(\mathbf{X}, \mathbf{A}), \tag{4}$$

where $\mathbf{X}$ is the node feature matrix, $\mathbf{I}$ denotes the identity matrix.

To enhance the discrepancy between view-specific predictions and avoid mutual interference between the different training objectives (further discussed in Section 4.2.2), distinct linear classifiers are employed for each view. The ego-view prediction matrix $\mathbf{P}_e \in \mathbb{R}^{|\mathcal{V}| \times C}$ and the structure-view prediction $\mathbf{P}_s \in \mathbb{R}^{|\mathcal{V}| \times C}$ are computed using classifiers $g_{\phi_1}$ and $g_{\phi_2}$, respectively:

$$\mathbf{P}_e = g_{\phi_1}(\mathbf{Z}_e), \quad \mathbf{P}_s = g_{\phi_2}(\mathbf{Z}_s). \tag{5}$$

The prediction discrepancy between the two views for node $i \in \mathcal{V}_{\text{train}}$ is quantified using the $\ell^2$-norm:

$$\delta_i = \left\| \mathbf{P}_e^{[i]} - \mathbf{P}_s^{[i]} \right\|_2, \tag{6}$$

where $\mathbf{P}_{\cdot}^{[i]}$ represents the prediction vector of node $i$.

To adapt flexibly across different graphs and noise scenarios, the discrepancy set $\mathcal{D} = \{\delta_i \mid i \in \mathcal{V}_{\text{train}}\}$ is modeled using a two-component Gaussian Mixture Model (GMM) via the Expectation-Maximization (EM) algorithm. Each discrepancy value $\delta_i$ is mapped to a posterior probability $\mathbb{P}_{\text{clean}}(s \mid \delta_i)$, indicating the likelihood of belonging to the clean-label component $s$ with the smaller mean. A threshold $\tau$ is then applied to partition the training set into clean and noisy subsets:

$$\mathcal{V}_{\text{clean}} = \{i \mid \mathbb{P}_{\text{clean}}(s \mid \delta_i) > \tau, i \in \mathcal{V}_{\text{train}}\}, \quad \mathcal{V}_{\text{noisy}} = \mathcal{V}_{\text{train}} \setminus \mathcal{V}_{\text{clean}}. \tag{7}$$

## 4.2 Model Training

### 4.2.1 Label Purification

After identifying nodes with noisy labels, pseudo-labels are generated to purify corrupted supervision. However, predictions obtained from either the ego-view or the structure-view exhibit inherent bias, as each is derived from a single source of information. To mitigate this issue and improve pseudo-label reliability, a mixed-view representation is introduced. The mixed-view integrates node features with local structural context to provide more accurate semantic estimates. Formally, the mixed-view representation is computed as:

$$\mathbf{Z}_m = f_\theta(\mathbf{X}, \tilde{\mathbf{A}}), \tag{8}$$

where $\tilde{\mathbf{A}}$ denotes the adjacency matrix with added self-loops. Representations are updated following the standard GCN formulation [16] as described in Eq. 2. Mixed-view predictions are then obtained via the classifier $g_{\phi_2}$ as $\mathbf{P}_m = g_{\phi_2}(\mathbf{Z}_m)$. To ensure pseudo-label quality, only high-confidence predictions are retained. Specifically, noisy nodes with prediction confidence exceeding a predefined threshold $\gamma$ are selected to construct the purified training set:

$$\mathcal{V}_{\text{purified}} = \{i \mid \max(\mathbf{P}_m^{[i]}) > \gamma, \ i \in \mathcal{V}_{\text{noisy}}\}. \tag{9}$$

For each node selected into $\mathcal{V}_{\text{purified}}$, the pseudo-label is determined by the class with the highest predicted probability:

$$\hat{\mathbf{Y}}^{[i]} = \arg\max \mathbf{P}_m^{[i]}, \quad i \in \mathcal{V}_{\text{purified}}. \tag{10}$$

### 4.2.2 Training Objective

As training progresses, predictions from different views tend to overfit noisy labels, resulting in a reduced discrepancy between views, especially for mislabeled nodes. The diminished discrepancy degrades the effectiveness of noise detection. To address this issue, view-specific training objectives are introduced to encourage distinct learning for each view. Specifically, the overall training objective

comprises three loss terms corresponding to the ego-view, structure-view, and mixed-view, each optimized on dedicated training nodes and labels.

First, the ego-view loss employs the original noisy labels $\tilde{\mathbf{Y}}$ to supervise predictions over all training nodes $\mathcal{V}_{\text{train}}$, allowing the model to closely capture potential label inconsistencies. Letting $L(\cdot, \cdot)$ denote the cross-entropy loss function, the ego-view loss is formulated as:

$$\mathcal{L}_e = \frac{1}{|\mathcal{V}_{\text{train}}|} \sum_{i \in \mathcal{V}_{\text{train}}} L(\mathbf{P}_e^{[i]}, \tilde{\mathbf{Y}}^{[i]}). \tag{11}$$

For the structure-view, pseudo-labels $\hat{\mathbf{Y}}$ generated from the mixed-view serve as the supervisory signals. The mixed-view integrates both node features and local structural context, producing more reliable pseudo-labels that mitigate the effect of noisy labels. Notably, nodes estimated to be mislabeled receive divergent supervision: the structure-view is trained with pseudo-labels, while the ego-view relies on the original noisy labels. This asymmetry amplifies the prediction discrepancy between views, which in turn facilitates the identification of mislabeled nodes. The structure-view loss is defined as:

$$\mathcal{L}_s = \frac{1}{|\mathcal{V}_{\text{clean}} \cup \mathcal{V}_{\text{purified}}|} \left( \sum_{i \in \mathcal{V}_{\text{clean}}} L(\mathbf{P}_s^{[i]}, \tilde{\mathbf{Y}}^{[i]}) + \sum_{i \in \mathcal{V}_{\text{purified}}} L(\mathbf{P}_s^{[i]}, \hat{\mathbf{Y}}^{[i]}) \right). \tag{12}$$

The mixed-view is trained using the original noisy labels $\tilde{\mathbf{Y}}$, but restricted to the confidently identified clean nodes $\mathcal{V}_{\text{clean}}$. This design aims to mitigate confirmation bias, which refers to the risk of reinforcing incorrect predictions when the model is trained on unreliable pseudo-labels. The mixed-view loss is given by:

$$\mathcal{L}_m = \frac{1}{|\mathcal{V}_{\text{clean}}|} \sum_{i \in \mathcal{V}_{\text{clean}}} L(\mathbf{P}_m^{[i]}, \tilde{\mathbf{Y}}^{[i]}). \tag{13}$$

Finally, these three losses are jointly optimized, resulting in the overall training objective:

$$\mathcal{L} = \mathcal{L}_e + \mathcal{L}_s + \mathcal{L}_m. \tag{14}$$

During the evaluation stage, predictions $\mathbf{P}_m$ from the mixed-view are employed as the final classification outputs.

## 5 Theoretical Justification

To theoretically justify the use of prediction discrepancy as an indicator of label noise, we analyze the deviation between ego-view and structure-view representations in graph neural networks. We begin by introducing assumptions on graphs and then propose a theorem that characterizes the probabilistic behavior of the discrepancy.

**Assumptions on Graphs.** Following previous works [22, 47], we consider a graph $\mathcal{G}$, where each node $i$ has features $\mathbf{x}_i \in \mathbb{R}^d$ and label $y_i$. We assume that (1) The features of node $i$ are sampled from feature distribution $\mathcal{F}_{y_i}$, i.e., $\mathbf{x}_i \sim \mathcal{F}_{y_i}$; (2) Dimensions of $\mathbf{x}_i$ are independent to each other; (3) The features in $\mathbf{X}$ are bounded by a positive scalar $B$, i.e., $\max_{i,j} |\mathbf{X}[i, j]| \leq B$; (4) For node $i$, its neighbor's labels are independently sampled from neighbor distribution $\mathcal{D}_{y_i}$. The sampling is repeated for $\deg(i)$ times to sample the labels for $\deg(i)$ neighbors.

We denote a graph following these assumptions (1)–(4) as $\mathcal{G} = \{\mathcal{V}, \mathcal{E}, \{\mathcal{F}_c, c \in \mathcal{C}\}, \{\mathcal{D}_c, c \in \mathcal{C}\}\}$. Note that we use the subscripts in $\mathcal{F}_{y_i}$ and $\mathcal{D}_{y_i}$ to indicate that these two distributions are shared by all nodes with the same label as node $i$. Define $\boldsymbol{\mu}_e := \mathbb{E}_{\mathbf{x} \sim \mathcal{F}_{y_i}}[\mathbf{x}]$ as the expected feature of node $i$ given the ground-truth label $y_i$, and define $\boldsymbol{\mu}_s := \mathbb{E}_{c \sim \mathcal{D}_{y_i}, \mathbf{x} \sim \mathcal{F}_c}[\mathbf{x}]$ as the expected feature of neighbors. We analyze the embeddings obtained after a GCN operation. Following previous works [18, 3, 22, 47], we drop the non-linearity in the analysis.

**Theorem 1.** *Consider a graph $\mathcal{G} = \{\mathcal{V}, \mathcal{E}, \{\mathcal{F}_c, c \in \mathcal{C}\}, \{\mathcal{D}_c, c \in \mathcal{C}\}\}$ that satisfies Assumptions (1)–(4). For any node $i \in \mathcal{V}$, let $\mathbf{h}_i = \mathbf{W}\mathbf{x}_i$ denote the pre-activation output of a single-layer GCN using only the node's own features (ego-view), where $\mathbf{x}_i \sim \mathcal{F}_{y_i}$. Let $\overline{\mathbf{h}}_i = \mathbf{W} \left( \frac{1}{|\mathcal{N}_i|} \sum_{j \in \mathcal{N}_i} \mathbf{x}_j \right)$ denote the corresponding structure-view representation that aggregates features from neighbors. Let*

Table 1: Node classification accuracy (%, mean ± std) on noisy datasets. $p$ denotes the noise rate for each setting. Best results are highlighted in **bold**, and second-best results are underlined.

| Dataset | Method | Uniform | | | Pair | | |
|---|---|---|---|---|---|---|---|
| | | $p = 0.2$ | $p = 0.4$ | $p = 0.6$ | $p = 0.2$ | $p = 0.3$ | $p = 0.4$ |
| Computer | GCN | $85.33 \pm 0.45$ | $82.68 \pm 0.66$ | $80.78 \pm 0.85$ | $84.11 \pm 0.51$ | $82.05 \pm 0.90$ | $77.85 \pm 1.89$ |
| | NRGNN | $86.86 \pm 0.67$ | $83.09 \pm 0.90$ | $80.04 \pm 4.16$ | $85.75 \pm 1.60$ | $83.59 \pm 0.80$ | $76.20 \pm 1.72$ |
| | RTGNN | $83.25 \pm 1.14$ | $82.44 \pm 1.63$ | $79.94 \pm 2.52$ | $82.54 \pm 0.65$ | $80.14 \pm 1.09$ | $72.24 \pm 1.92$ |
| | PI-GNN | $83.25 \pm 1.94$ | $81.00 \pm 1.89$ | $79.10 \pm 2.24$ | $81.42 \pm 2.41$ | $79.81 \pm 1.03$ | $75.80 \pm 2.89$ |
| | TSS | $\underline{86.87 \pm 2.14}$ | $83.62 \pm 1.73$ | $82.11 \pm 1.02$ | $83.80 \pm 1.78$ | $82.56 \pm 1.75$ | $79.89 \pm 2.03$ |
| | ERASE | $86.77 \pm 0.93$ | $\underline{84.84 \pm 0.83}$ | $\underline{83.41 \pm 1.13}$ | $\underline{86.68 \pm 0.75}$ | $\underline{84.60 \pm 1.03}$ | $\underline{80.13 \pm 4.42}$ |
| | GD$^2$ | $\mathbf{86.91 \pm 0.56}$ | $\mathbf{86.26 \pm 0.71}$ | $\mathbf{84.76 \pm 0.81}$ | $\mathbf{86.88 \pm 0.55}$ | $\mathbf{85.75 \pm 0.64}$ | $\mathbf{83.16 \pm 1.75}$ |
| Photo | GCN | $90.10 \pm 0.72$ | $86.38 \pm 1.27$ | $72.55 \pm 5.13$ | $89.03 \pm 0.49$ | $87.26 \pm 0.78$ | $85.84 \pm 2.21$ |
| | NRGNN | $89.29 \pm 1.99$ | $83.25 \pm 5.20$ | $69.64 \pm 6.85$ | $88.36 \pm 1.21$ | $83.10 \pm 3.94$ | $82.16 \pm 5.35$ |
| | RTGNN | $89.65 \pm 1.10$ | $87.35 \pm 2.58$ | $73.49 \pm 3.38$ | $\underline{91.66 \pm 1.06}$ | $89.90 \pm 1.89$ | $87.03 \pm 6.18$ |
| | PI-GNN | $90.83 \pm 0.71$ | $87.07 \pm 0.81$ | $73.33 \pm 1.44$ | $\underline{89.58 \pm 0.41}$ | $88.18 \pm 1.54$ | $\underline{87.79 \pm 4.81}$ |
| | TSS | $91.66 \pm 1.08$ | $87.27 \pm 0.95$ | $73.07 \pm 4.46$ | $89.03 \pm 0.85$ | $88.00 \pm 2.13$ | $86.04 \pm 5.75$ |
| | ERASE | $\underline{91.86 \pm 0.47}$ | $\underline{88.00 \pm 0.53}$ | $\mathbf{75.10 \pm 1.57}$ | $91.41 \pm 0.75$ | $87.67 \pm 1.47$ | $85.59 \pm 2.33$ |
| | GD$^2$ | $\mathbf{92.45 \pm 1.30}$ | $\mathbf{89.72 \pm 0.98}$ | $\underline{74.32 \pm 5.85}$ | $\mathbf{92.83 \pm 1.10}$ | $\mathbf{91.84 \pm 1.00}$ | $\mathbf{89.00 \pm 1.23}$ |
| CS | GCN | $91.51 \pm 0.13$ | $90.57 \pm 0.29$ | $88.07 \pm 0.72$ | $91.07 \pm 0.27$ | $88.54 \pm 0.46$ | $80.67 \pm 2.12$ |
| | NRGNN | $92.07 \pm 0.50$ | $91.40 \pm 0.46$ | $89.37 \pm 1.23$ | $89.50 \pm 0.14$ | $86.44 \pm 1.10$ | $75.51 \pm 6.00$ |
| | RTGNN | $\underline{93.23 \pm 0.33}$ | $91.91 \pm 0.55$ | $\mathbf{90.62 \pm 0.94}$ | $\underline{92.44 \pm 0.52}$ | $89.69 \pm 1.33$ | $76.16 \pm 8.44$ |
| | PI-GNN | $92.40 \pm 0.18$ | $91.68 \pm 0.34$ | $89.57 \pm 0.88$ | $91.17 \pm 0.76$ | $\underline{89.75 \pm 0.93}$ | $\underline{82.68 \pm 5.65}$ |
| | TSS | $89.80 \pm 0.52$ | $87.23 \pm 1.04$ | $82.13 \pm 1.71$ | $87.70 \pm 0.94$ | $84.69 \pm 2.01$ | $75.83 \pm 7.20$ |
| | ERASE | $91.52 \pm 0.25$ | $91.18 \pm 0.33$ | $89.00 \pm 0.46$ | $90.23 \pm 0.35$ | $86.92 \pm 1.34$ | $81.66 \pm 2.66$ |
| | GD$^2$ | $\mathbf{93.43 \pm 0.34}$ | $\mathbf{92.37 \pm 0.45}$ | $\underline{89.61 \pm 0.54}$ | $\mathbf{92.75 \pm 0.38}$ | $\mathbf{90.54 \pm 0.43}$ | $\mathbf{84.68 \pm 1.92}$ |
| WikiCS | GCN | $78.35 \pm 0.49$ | $75.06 \pm 0.52$ | $72.36 \pm 0.86$ | $78.84 \pm 0.67$ | $74.33 \pm 1.00$ | $69.15 \pm 1.75$ |
| | NRGNN | $74.52 \pm 0.97$ | $69.48 \pm 1.16$ | $64.73 \pm 2.81$ | $74.89 \pm 1.66$ | $71.01 \pm 1.65$ | $67.74 \pm 2.83$ |
| | RTGNN | $76.75 \pm 0.57$ | $75.46 \pm 0.70$ | $72.35 \pm 1.44$ | $76.68 \pm 0.67$ | $\underline{75.12 \pm 1.93}$ | $71.56 \pm 3.47$ |
| | PI-GNN | $74.11 \pm 1.66$ | $71.48 \pm 1.33$ | $69.85 \pm 6.87$ | $65.05 \pm 8.78$ | $63.40 \pm 7.91$ | $\underline{59.28 \pm 8.92}$ |
| | TSS | $78.23 \pm 0.28$ | $\underline{77.39 \pm 0.66}$ | $69.98 \pm 0.85$ | $77.76 \pm 0.40$ | $74.67 \pm 1.03$ | $65.43 \pm 3.49$ |
| | ERASE | $\mathbf{79.85 \pm 0.62}$ | $77.27 \pm 0.66$ | $\underline{74.08 \pm 0.76}$ | $\underline{78.86 \pm 0.52}$ | $74.43 \pm 1.52$ | $69.74 \pm 4.07$ |
| | GD$^2$ | $\underline{79.45 \pm 0.36}$ | $\mathbf{78.21 \pm 0.60}$ | $\mathbf{74.47 \pm 0.80}$ | $\mathbf{79.84 \pm 0.29}$ | $\mathbf{78.09 \pm 0.42}$ | $\mathbf{75.88 \pm 1.68}$ |
| Roman-Empire | GCN | $78.35 \pm 0.49$ | $57.86 \pm 2.00$ | $50.50 \pm 1.02$ | $\underline{66.78 \pm 1.05}$ | $64.16 \pm 1.04$ | $58.34 \pm 2.62$ |
| | NRGNN | $61.85 \pm 0.46$ | $59.53 \pm 1.01$ | $49.60 \pm 1.36$ | $62.41 \pm 0.68$ | $60.55 \pm 0.69$ | $53.71 \pm 1.39$ |
| | RTGNN | $\underline{64.85 \pm 0.47}$ | $60.13 \pm 0.98$ | $\mathbf{51.80 \pm 1.31}$ | $62.28 \pm 0.71$ | $59.82 \pm 0.92$ | $51.61 \pm 1.68$ |
| | PI-GNN | $63.64 \pm 0.48$ | $\underline{57.49 \pm 1.51}$ | $47.74 \pm 2.15$ | $62.45 \pm 0.99$ | $57.87 \pm 1.32$ | $50.06 \pm 1.81$ |
| | TSS | $58.29 \pm 0.35$ | $54.18 \pm 0.66$ | $47.66 \pm 1.10$ | $66.35 \pm 0.48$ | $\underline{63.60 \pm 0.52}$ | $\underline{58.52 \pm 1.04}$ |
| | ERASE | $63.26 \pm 0.41$ | $51.66 \pm 0.31$ | $48.74 \pm 0.44$ | $61.53 \pm 0.76$ | $57.42 \pm 1.48$ | $51.81 \pm 2.27$ |
| | GD$^2$ | $\mathbf{66.19 \pm 0.69}$ | $\mathbf{60.26 \pm 0.80}$ | $\underline{51.24 \pm 1.68}$ | $\mathbf{68.25 \pm 1.01}$ | $\mathbf{66.22 \pm 0.86}$ | $\mathbf{62.16 \pm 1.14}$ |
| Amazon-Ratings | GCN | $36.82 \pm 0.65$ | $35.85 \pm 2.16$ | $34.95 \pm 2.10$ | $37.58 \pm 1.07$ | $35.09 \pm 3.06$ | $32.99 \pm 2.12$ |
| | NRGNN | $37.90 \pm 0.26$ | $37.03 \pm 0.84$ | $35.33 \pm 1.06$ | $37.15 \pm 1.32$ | $36.98 \pm 0.62$ | $\underline{36.52 \pm 0.85}$ |
| | RTGNN | $36.65 \pm 0.01$ | $36.62 \pm 0.04$ | $\underline{34.59 \pm 0.18}$ | $36.59 \pm 0.12$ | $34.58 \pm 3.64$ | $30.87 \pm 3.39$ |
| | PI-GNN | $37.22 \pm 0.42$ | $35.10 \pm 0.80$ | $33.17 \pm 2.25$ | $36.60 \pm 0.49$ | $33.35 \pm 1.42$ | $31.98 \pm 2.73$ |
| | TSS | $\underline{39.56 \pm 0.55}$ | $\mathbf{37.62 \pm 0.97}$ | $34.73 \pm 2.84$ | $\underline{39.65 \pm 0.52}$ | $\mathbf{38.99 \pm 0.84}$ | $36.45 \pm 1.46$ |
| | ERASE | $38.85 \pm 0.62$ | $36.53 \pm 0.48$ | $33.26 \pm 0.74$ | $37.11 \pm 1.99$ | $35.47 \pm 2.87$ | $31.67 \pm 5.13$ |
| | GD$^2$ | $\mathbf{39.80 \pm 0.73}$ | $\underline{37.58 \pm 1.10}$ | $\mathbf{35.72 \pm 1.04}$ | $\mathbf{39.97 \pm 0.55}$ | $\underline{38.76 \pm 0.74}$ | $\mathbf{38.28 \pm 0.90}$ |

$d$ denote the feature dimensionality, and $\rho(\mathbf{W})$ denote the largest singular value of the weight matrix $\mathbf{W}$. Define the discrepancy between the two views as $\delta_i = \|\mathbf{h}_i - \bar{\mathbf{h}}_i\|_2$. Then, for any $t > 0$, the probability that $\delta_i$ exceeds $t$ is bounded by

$$\mathbb{P}(\delta_i \geq t) \leq 2d \cdot \exp\left(-\frac{\deg(i)+1}{2B^2d}\left(\frac{t}{\rho(\mathbf{W})} - \|\boldsymbol{\mu}_e - \boldsymbol{\mu}_s\|_2\right)^2\right). \tag{15}$$

The proof is provided in Appendix D. Theorem 1 shows that clean-label nodes inherently exhibit bounded representation discrepancies between the ego-view and structure-view. This behavior arises from the statistical alignment between node-specific features $\boldsymbol{\mu}_e$ and neighborhood semantics $\boldsymbol{\mu}_s$. Furthermore, the bound strengthens with increasing node degree $\deg(i)$, implying that nodes with richer neighborhood contexts exhibit more stable discrepancy patterns. This explains the robustness of structure-view representations under label noise, as neighborhood aggregation smooths out individual perturbations. In contrast, the ego-view representation of a mislabeled node is constructed independently of neighbor information and is therefore more susceptible to corrupted supervision. As a result, the representation deviates from the structure-view and violates the concentration behavior observed in clean-label cases.

Table 2: Ablation of $GD^2$ on the Computer and Roman-Empire datasets under the Pair-0.4 noise.

| Ablation | $\mathcal{L}_e$ | $\mathcal{L}_s$ | Separate classifier | Computer | Roman-Empire |
|---|---|---|---|---|---|
| $GD^2$ | $\mathcal{V}_{\text{train}}$ | $\mathcal{V}_{\text{clean}} \cup \mathcal{V}_{\text{purified}}$ | $\checkmark$ | **83.16 $\pm$ 1.75** | **62.16 $\pm$ 1.14** |
| $GD^2$ with $\phi_1 \equiv \phi_2$ | $\mathcal{V}_{\text{train}}$ | $\mathcal{V}_{\text{clean}} \cup \mathcal{V}_{\text{purified}}$ | $\times$ | 81.89 $\pm$ 1.76 | 60.87 $\pm$ 2.52 |
| $GD^2$ w/o label purification | $\mathcal{V}_{\text{train}}$ | $\mathcal{V}_{\text{clean}}$ | $\checkmark$ | 82.35 $\pm$ 1.29 | 60.76 $\pm$ 1.70 |
| $GD^2$ w/o distinct objectives | $\mathcal{V}_{\text{train}}$ $\mathcal{V}_{\text{clean}}$ $\mathcal{V}_{\text{clean}} \cup \mathcal{V}_{\text{purified}}$ | $\mathcal{V}_{\text{train}}$ $\mathcal{V}_{\text{clean}}$ $\mathcal{V}_{\text{clean}} \cup \mathcal{V}_{\text{purified}}$ | $\checkmark$ $\checkmark$ $\checkmark$ | 81.13 $\pm$ 1.26 82.69 $\pm$ 0.88 82.35 $\pm$ 2.06 | 61.62 $\pm$ 1.48 60.97 $\pm$ 2.73 61.13 $\pm$ 1.06 |

# 6 Experiments

## 6.1 Setup

**Datasets and Settings.** We conduct experiments on six benchmark datasets, including four homophilous graphs: Computer, Photo, CS [29], and WikiCS [25], and two heterophilous graphs: Roman-Empire, and Amazon-Ratings [26]. Dataset statistics are provided in Appendix A. Following previous works [12, 48], we randomly select 10% of the nodes for training, 10% for validation, and use the remaining nodes for testing. All results are averaged over 10 runs with different random seeds, and we report both the mean and standard deviation.

Following prior studies [5, 2, 28, 8], we synthetically corrupt labels of the training and validation sets. We consider two types of label noise:

**1)** *Uniform-$p$*: Each label is flipped to a randomly selected incorrect class with probability $p$.

**2)** *Pair-$p$*: Each label is flipped to a corresponding pair class with probability $p$, simulating realistic mislabeling patterns.

We set $p \in \{0.2, 0.4, 0.6\}$ for uniform noise and $p \in \{0.2, 0.3, 0.4\}$ for pair noise. More implementation details, including model architectures and training configurations, are provided in Appendix B.

**Baselines.** We compare $GD^2$ with several representative methods designed for learning with noisy labels on graph data. The baselines include: (1) NRGNN [5], which refines the graph structure to alleviate the effects of label noise; (2) ERASE [2], which performs label denoising via label propagation; (3) RTGNN [28] and PI-GNN [8], which incorporate auxiliary supervision; and (4) TSS [39], which leverages graph topology for node selection under a curriculum learning framework. All hyperparameters are tuned following the original publications. Additional comparisons with general noisy label learning methods are provided in Appendix C.

## 6.2 Main Results

Table 1 reports the node classification performance across diverse noise settings and datasets. As shown in the table, $GD^2$ achieves the best or second-best performance across a wide range of datasets and noise settings. Compared to representative graph-specific methods such as NRGNN and ERASE, $GD^2$ achieves substantial performance improvements, particularly under high noise rates. For example, on Computer, $GD^2$ outperforms ERASE by more than *3.03%* under pair-0.4 noise. These consistent improvements indicate that $GD^2$ offers a general solution for robust node classification under label noise, delivering strong performance across diverse graphs and noise settings.

## 6.3 Ablation Study

**Effect of separate classifiers.** We begin by evaluating the necessity of using separate classifiers for ego-view and structure-view representations. The variant *$GD^2$ with $\phi_1 \equiv \phi_2$* employs a shared classifier head for both views while keeping all other settings unchanged. As shown in Table 2, this modification results in a noticeable performance drop ($-1.27\%$ on Computer and $-1.29\%$ on Roman-Empire), indicating that separate classifiers are crucial for preventing interference between views. The decoupled heads allow each view to independently learn view-specific predictions and avoid mutual supervision contamination.

**Effect of label purification.** To evaluate the effect of label purification, we compare $GD^2$ with the variant *$GD^2$ w/o label purification*, where $\mathcal{L}_s$ is trained only on the clean set while the purified

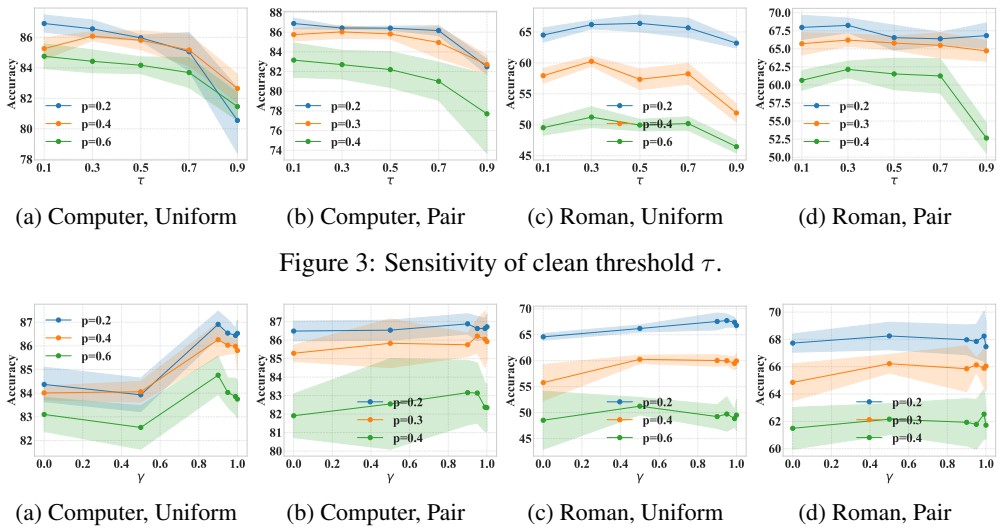

Figure 3: Sensitivity of clean threshold $\tau$.

(a) Computer, Uniform  (b) Computer, Pair  (c) Roman, Uniform  (d) Roman, Pair

(a) Computer, Uniform  (b) Computer, Pair  (c) Roman, Uniform  (d) Roman, Pair

Figure 4: Sensitivity of pseudo-label confidence threshold $\gamma$.

set is excluded. As reported in Table 2, this variant performs worse than $GD^2$, confirming that label purification is beneficial not only for supervision but also for enhancing the prediction discrepancy signal. Although the purified set is not directly used for training the mix-view branch, it plays an essential role in amplifying cross-view disagreement, thereby improving noise detection quality.

**Effect of distinct training objectives.** Finally, we assess the impact of using distinct training objectives for the ego and structure views. The variant *GD² w/o distinct objectives* uses the same training set for both $\mathcal{L}_e$ and $\mathcal{L}_s$, with three configurations considered: (i) both views are trained on the full noisy set $\mathcal{V}_{\text{train}}$, (ii) both are trained on the clean set $\mathcal{V}_{\text{clean}}$, and (iii) both are trained on the union of clean and purified nodes $\mathcal{V}_{\text{clean}} \cup \mathcal{V}_{\text{purified}}$. As shown in Table 2, all three variants exhibit performance degradation compared to $GD^2$, with the largest drop reaching $2.03\%$ on Computer and $1.19\%$ on Roman-Empire. These results highlight the importance of assigning different supervision to the views, which amplify prediction discrepancies among views.

### 6.4 Hyperparameter Sensitivity

**Effect of clean threshold $\tau$.** The clean threshold $\tau$ determines the minimum agreement required between ego-view and structure-view predictions for a node to be considered clean. Figure 3 shows the accuracy of $GD^2$ as $\tau$ varies from 0.1 to 0.9. Overall, increasing $\tau$ leads to a gradual decline in performance, as stricter thresholds reduce the coverage of clean samples. However, when $\tau \leq 0.7$, the degradation remains limited, suggesting that $GD^2$ is relatively robust to moderate threshold choices. A sharp performance drop is observed at $\tau = 0.9$, indicating that overly conservative filtering discards too much supervision and impairs learning. These results highlight the importance of avoiding excessively large thresholds, while also demonstrating stability under moderate settings.

**Effect of pseudo-label confidence threshold $\gamma$.** The pseudo-label confidence threshold $\gamma$ controls the trade-off between the quantity and quality of pseudo-labels used in label purification. Figure 4 reports the accuracy of $GD^2$ as $\gamma$ varies from 0.0 to 1.0. As $\gamma$ increases, accuracy generally improves, suggesting that high-confidence pseudo-labels provide more reliable supervision and enhance model robustness. In the low-$\gamma$ regime, performance exhibits greater variance and tends to degrade. In contrast, when $\gamma$ is sufficiently large (e.g., $\gamma \geq 0.9$), accuracy becomes both higher and more stable, indicating that $GD^2$ is relatively insensitive to the threshold within the high-confidence range. Notably, setting $\gamma = 1.0$, which disables pseudo-labeling entirely, results in inferior performance compared to slightly relaxed thresholds (e.g., $\gamma = 0.9$). This confirms that incorporating a small number of highly confident pseudo-labels is more beneficial than discarding them altogether. These results highlight the importance of maintaining both precision and minimal coverage in the label purification process.

# 7 Conclusion

In this paper, we propose a novel graph learning framework $GD^2$ to mitigate the detrimental effects of label noise in Graph Neural Networks (GNNs). Our key insight is leveraging the prediction discrepancy derived from two complementary views—the ego-view, capturing individual node semantics, and the structure-view, encoding neighborhood structural semantics. This discrepancy serves as an intrinsic signal for noisy-label detection. Furthermore, we introduce a view-specific training strategy that amplifies prediction discrepancies through differentiated supervision, enhancing the noise detection module. Extensive experiments across multiple real-world datasets under various noise conditions demonstrated that $GD^2$ outperforms state-of-the-art noise-robust GNN baselines. These results demonstrate the effectiveness of modeling node–neighbor dependencies for label noise detection and highlight the potential of graph-specific relational patterns in improving robustness. Future work includes extending this framework to dynamic and heterogeneous graphs, where structural variations and multiple relation types pose additional challenges for noise detection.

## Acknowledgments and Disclosure of Funding

This work was supported in part by National Key R&D Program of China No. 2024YFB2705300, NSFC Grant 62232011, 62402315, and the Shanghai Science and Technology Innovation Action Plan Grant 24BC3201200.

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

## A  Dataset Statistics

Table 3: Statistics of used graph datasets in this paper.

| Dataset | # Nodes | # Edges | # Classes | # Features | Type |
|---------|---------|---------|-----------|------------|------|
| Computer | 13,752 | 245,861 | 10 | 767 | Homophily |
| Photo | 7,650 | 119,081 | 8 | 745 | Homophily |
| CS | 18,333 | 81,894 | 15 | 6,805 | Homophily |
| WikiCS | 11,701 | 431,206 | 10 | 300 | Homophily |
| Roman-Empire | 22,662 | 32,927 | 18 | 300 | Heterophily |
| Amazon-Ratings | 24,492 | 93,050 | 5 | 300 | Heterophily |

## B  Implementation Details

The GNN encoder $f_\theta$ is implemented as a Graph Convolutional Network (GCN) optionally equipped with residual connections, Batch Normalization, or Layer Normalization, depending on the specific dataset. The classifiers $g_{\phi_1}$ and $g_{\phi_2}$ are implemented as single-layer linear networks. We use the AdamW optimizer for training. For graph data augmentations, we adopt a combination of random node feature masking and edge masking, where node features and edges are independently masked with probabilities $p_f$ and $p_e$, respectively.

**Hyperparameters.**  We select the hyperparameters through a grid search, with the search ranges detailed in Table 4. All hyperparameters are tuned using validation sets. For baseline methods, hyperparameter tuning is conducted within the ranges recommended in their original publications.

Table 4: Search ranges for hyperparameters.

| Hyperparameter | Search range |
|----------------|--------------|
| Traing epoch | 500, 1000 |
| Learning rate | 0.001, 0.005, 0.01 |
| Hidden dim | 64, 256, 512 |
| GNN Layers | 1, 2, 3, 4, 5 |
| Normalization | Batch Norm, Layer Norm |
| Residual Connection | True, False |
| Dropout | 0.2, 0.3, 0.5, 0.7 |
| Feature masking prob. | 0.0, 0.1, 0.2, 0.3, 0.4 |
| Edge dropping prob. | 0.0, 0.1, 0.2, 0.3, 0.4 |
| Clean prob. threshold $\tau$ | 0.1, 0.3, 0.5, 0.7, 0.9 |
| Pseudo-label confidence threshold $\gamma$ | 0.0, 0.5, 0.9, 0.95, 0.99, 1.0 |

**Label Noise Generation**  Following prior work [5, 8, 30], label noise is simulated by corrupting the ground-truth labels. The noise generation process is characterized using a noise transition matrix $Q \in \mathbb{R}^{C \times C}$, where $C$ denotes the number of classes. Each entry $Q_{ij} = \Pr(y_{\text{noisy}} = j \mid y_{\text{true}} = i)$ represents the probability of a true label $i$ being flipped to a noisy label $j$.

For Uniform-$p$ noise, the transition matrix $Q$ takes the following form:

$$
Q = \begin{bmatrix}
1-p & \frac{p}{C-1} & \cdots & \frac{p}{C-1} \\
\frac{p}{C-1} & 1-p & \cdots & \frac{p}{C-1} \\
\vdots & \vdots & \ddots & \vdots \\
\frac{p}{C-1} & \frac{p}{C-1} & \cdots & 1-p
\end{bmatrix}
$$

For Pair-$p$ noise, a typical transition matrix $Q$ is defined as:

$$
Q = \begin{bmatrix}
1-p & p & 0 & \cdots & 0 \\
0 & 1-p & p & \cdots & 0 \\
\vdots & \ddots & \ddots & \ddots & \vdots \\
0 & \cdots & 0 & 1-p & p \\
p & 0 & \cdots & 0 & 1-p
\end{bmatrix}
$$

Note that $GD^2$ does not rely on knowledge of the transition matrix or the noise rate to operate effectively.

**Infrastructures.** We implement the proposed $GD^2$ with PyTorch 2.4.0 and torch_geometric 2.6.1. All experiments are conducted on a Linux server with a NVIDIA GeForce RTX 4090 GPU with 24GB memory.

## C   Performance Comparison against General Noisy Label Learning Methods

We further compare $GD^2$ with representative general noisy label learning methods that are originally designed for i.i.d. data rather than graph-structured data. The baselines include: (i) robust loss function approaches such as SCE [31] and APL [21]; (ii) small-loss based sample selection methods such as Co-teaching [13] and JoCoR [33]; and (iii) confidence-based sample selection and semi-supervised learning methods, exemplified by ProMix [41]. These methods represent different paradigms of noise-robust learning and provide a complementary evaluation to validate the effectiveness of $GD^2$.

Table 5 presents the comparison results. It can be observed that general noisy label learning methods, although effective on i.i.d. data, exhibit limited robustness when applied to graph-structured data. In contrast, $GD^2$ achieves consistently superior performance, demonstrating its effectiveness in addressing label noise by explicitly leveraging the structural information inherent in graphs. These results highlight the necessity of designing noise-robust methods tailored to graph domains rather than directly transferring techniques developed for i.i.d. settings.

## D   Proof of Theorem

*Proof.* The expectation of $\mathbf{h}_i$ is given by:

$$
\mathbb{E}[\mathbf{h}_i] = \mathbb{E}[\mathbf{W}\mathbf{x}_i] = \mathbf{W}\mathbb{E}[\mathbf{x}_i] = \mathbf{W}\boldsymbol{\mu}_e,
$$

and the aggregated neighbor representation is:

$$
\overline{\mathbf{h}}_i = \mathbf{W}\left(\frac{1}{|\mathcal{N}_i|}\sum_{j\in\mathcal{N}_i}\mathbf{x}_j\right), \quad \mathbb{E}[\overline{\mathbf{h}}_i] = \mathbf{W}\boldsymbol{\mu}_s.
$$

Define the discrepancy:

$$
\delta_i = \left\|\mathbf{h}_i - \overline{\mathbf{h}}_i\right\|_2 = \left\|\mathbf{W}\left(\mathbf{x}_i - \frac{1}{|\mathcal{N}_i|}\sum_{j\in\mathcal{N}_i}\mathbf{x}_j\right)\right\|_2 = \|\mathbf{W}\boldsymbol{\xi}_i\|_2,
$$

where we let:

$$
\boldsymbol{\xi}_i := \mathbf{x}_i - \frac{1}{|\mathcal{N}_i|}\sum_{j\in\mathcal{N}_i}\mathbf{x}_j.
$$

By the sub-multiplicativity of matrix norms, we have:

$$
\delta_i \leq \|\mathbf{W}\|_2 \cdot \|\boldsymbol{\xi}_i\|_2 = \rho(\mathbf{W}) \cdot \|\boldsymbol{\xi}_i\|_2.
$$

Thus, for any $t > 0$,

$$
\mathbb{P}(\delta_i \geq t) \leq \mathbb{P}\left(\|\boldsymbol{\xi}_i\|_2 \geq \frac{t}{\rho(\mathbf{W})}\right).
$$

Table 5: Node classification accuracy (%, mean ± std) on noisy datasets. $p$ denotes the noise rate for each setting. Best results are highlighted in **bold**.

| Dataset | Method | Uniform | | | Pair | | |
| --- | --- | --- | --- | --- | --- | --- | --- |
| | | $p = 0.2$ | $p = 0.4$ | $p = 0.6$ | $p = 0.2$ | $p = 0.3$ | $p = 0.4$ |
| Computer | SCE | $86.15 \pm 1.80$ | $84.41 \pm 0.92$ | $79.91 \pm 2.10$ | $86.05 \pm 0.36$ | $83.57 \pm 1.31$ | $76.51 \pm 2.16$ |
| | APL | $83.86 \pm 5.95$ | $79.22 \pm 2.25$ | $78.80 \pm 2.29$ | $77.91 \pm 3.14$ | $76.90 \pm 5.62$ | $74.95 \pm 4.51$ |
| | Co-teaching | $84.57 \pm 1.33$ | $83.59 \pm 1.38$ | $78.18 \pm 2.15$ | $82.62 \pm 1.16$ | $82.22 \pm 1.08$ | $74.38 \pm 1.47$ |
| | JoCoR | $81.10 \pm 1.29$ | $81.03 \pm 1.20$ | $78.14 \pm 1.02$ | $79.24 \pm 0.26$ | $77.57 \pm 0.78$ | $68.63 \pm 5.16$ |
| | ProMix | $84.66 \pm 1.01$ | $82.26 \pm 1.06$ | $80.61 \pm 1.52$ | $81.60 \pm 0.75$ | $80.11 \pm 0.92$ | $78.57 \pm 3.04$ |
| | GD$^2$ | $\mathbf{86.91 \pm 0.56}$ | $\mathbf{86.26 \pm 0.71}$ | $\mathbf{84.76 \pm 0.81}$ | $\mathbf{86.88 \pm 0.55}$ | $\mathbf{85.75 \pm 0.64}$ | $\mathbf{83.16 \pm 1.75}$ |
| Photo | SCE | $91.20 \pm 0.64$ | $88.86 \pm 0.93$ | $73.54 \pm 1.21$ | $89.90 \pm 1.08$ | $86.79 \pm 1.51$ | $76.21 \pm 3.54$ |
| | APL | $91.72 \pm 1.05$ | $89.73 \pm 0.73$ | $72.09 \pm 2.05$ | $89.80 \pm 2.55$ | $88.30 \pm 1.11$ | $83.05 \pm 3.06$ |
| | Co-teaching | $85.60 \pm 7.02$ | $77.11 \pm 6.62$ | $61.98 \pm 5.02$ | $86.43 \pm 5.64$ | $84.67 \pm 5.67$ | $78.78 \pm 9.57$ |
| | JoCoR | $88.69 \pm 3.36$ | $83.33 \pm 4.87$ | $68.32 \pm 4.67$ | $83.30 \pm 3.62$ | $81.41 \pm 2.96$ | $74.49 \pm 2.42$ |
| | ProMix | $91.76 \pm 1.83$ | $88.36 \pm 0.11$ | $68.59 \pm 2.40$ | $89.03 \pm 2.14$ | $88.60 \pm 1.11$ | $83.86 \pm 2.39$ |
| | GD$^2$ | $\mathbf{92.45 \pm 1.30}$ | $\mathbf{89.72 \pm 0.98}$ | $\mathbf{74.32 \pm 5.85}$ | $\mathbf{92.83 \pm 1.10}$ | $\mathbf{91.84 \pm 1.00}$ | $\mathbf{89.00 \pm 1.23}$ |
| CS | SCE | $92.12 \pm 0.29$ | $91.25 \pm 0.39$ | $88.22 \pm 1.38$ | $90.10 \pm 0.66$ | $87.23 \pm 1.53$ | $75.89 \pm 2.60$ |
| | APL | $92.18 \pm 0.22$ | $90.48 \pm 0.96$ | $87.00 \pm 0.56$ | $90.04 \pm 0.89$ | $86.74 \pm 1.90$ | $79.73 \pm 5.30$ |
| | Co-teaching | $91.66 \pm 0.37$ | $90.52 \pm 0.48$ | $86.38 \pm 1.96$ | $90.17 \pm 0.89$ | $87.26 \pm 0.80$ | $73.81 \pm 6.66$ |
| | JoCoR | $91.30 \pm 0.82$ | $91.17 \pm 0.34$ | $87.44 \pm 1.05$ | $88.45 \pm 1.65$ | $84.15 \pm 0.88$ | $70.32 \pm 6.13$ |
| | ProMix | $92.56 \pm 0.51$ | $90.33 \pm 0.59$ | $88.91 \pm 1.26$ | $90.98 \pm 0.61$ | $87.83 \pm 1.13$ | $77.31 \pm 3.37$ |
| | GD$^2$ | $\mathbf{93.43 \pm 0.34}$ | $\mathbf{92.37 \pm 0.45}$ | $\mathbf{89.61 \pm 0.54}$ | $\mathbf{92.75 \pm 0.38}$ | $\mathbf{90.54 \pm 0.43}$ | $\mathbf{84.68 \pm 1.92}$ |
| WikiCS | SCE | $72.21 \pm 0.63$ | $70.68 \pm 0.47$ | $61.89 \pm 1.94$ | $71.22 \pm 0.70$ | $68.90 \pm 0.85$ | $60.48 \pm 3.16$ |
| | APL | $71.61 \pm 2.16$ | $68.36 \pm 2.84$ | $60.82 \pm 2.17$ | $69.90 \pm 0.45$ | $69.09 \pm 0.73$ | $63.21 \pm 5.68$ |
| | Co-teaching | $76.15 \pm 1.46$ | $71.69 \pm 2.97$ | $60.65 \pm 1.58$ | $74.45 \pm 1.54$ | $71.53 \pm 1.81$ | $62.23 \pm 5.45$ |
| | JoCoR | $76.09 \pm 1.70$ | $74.84 \pm 1.90$ | $65.19 \pm 4.33$ | $73.50 \pm 0.79$ | $68.79 \pm 2.57$ | $60.39 \pm 3.32$ |
| | ProMix | $73.74 \pm 2.29$ | $72.52 \pm 2.81$ | $68.69 \pm 2.19$ | $74.45 \pm 0.85$ | $72.42 \pm 0.23$ | $67.69 \pm 5.71$ |
| | GD$^2$ | $\mathbf{79.45 \pm 0.36}$ | $\mathbf{78.21 \pm 0.60}$ | $\mathbf{74.47 \pm 0.80}$ | $\mathbf{79.84 \pm 0.29}$ | $\mathbf{78.09 \pm 0.42}$ | $\mathbf{75.88 \pm 1.68}$ |
| Roman-Empire | SCE | $57.78 \pm 1.13$ | $54.83 \pm 1.86$ | $50.46 \pm 3.35$ | $66.84 \pm 1.22$ | $54.14 \pm 0.60$ | $52.20 \pm 0.78$ |
| | APL | $63.29 \pm 1.46$ | $50.67 \pm 1.09$ | $45.82 \pm 2.61$ | $61.49 \pm 2.31$ | $58.29 \pm 0.98$ | $54.65 \pm 1.55$ |
| | Co-teaching | $53.92 \pm 0.43$ | $52.16 \pm 0.46$ | $48.53 \pm 0.82$ | $61.59 \pm 0.63$ | $59.13 \pm 0.57$ | $55.47 \pm 0.62$ |
| | JoCoR | $59.52 \pm 0.71$ | $55.25 \pm 0.75$ | $50.41 \pm 1.05$ | $66.91 \pm 0.22$ | $57.60 \pm 0.78$ | $53.67 \pm 1.70$ |
| | ProMix | $53.51 \pm 0.17$ | $52.27 \pm 0.37$ | $48.90 \pm 0.83$ | $61.68 \pm 0.80$ | $59.43 \pm 0.88$ | $55.50 \pm 0.49$ |
| | GD$^2$ | $\mathbf{66.19 \pm 0.69}$ | $\mathbf{60.26 \pm 0.80}$ | $\mathbf{51.24 \pm 1.68}$ | $\mathbf{68.25 \pm 1.01}$ | $\mathbf{66.22 \pm 0.86}$ | $\mathbf{62.16 \pm 1.14}$ |
| Amazon-Ratings | SCE | $36.66 \pm 0.17$ | $36.52 \pm 0.14$ | $36.29 \pm 0.33$ | $36.38 \pm 0.77$ | $37.25 \pm 0.40$ | $35.32 \pm 2.09$ |
| | APL | $38.28 \pm 1.49$ | $35.85 \pm 0.73$ | $34.88 \pm 2.19$ | $37.70 \pm 0.26$ | $36.89 \pm 0.75$ | $34.97 \pm 0.80$ |
| | Co-teaching | $38.86 \pm 1.91$ | $36.12 \pm 1.46$ | $35.02 \pm 1.00$ | $38.63 \pm 1.64$ | $37.71 \pm 0.75$ | $35.87 \pm 1.09$ |
| | JoCoR | $38.15 \pm 0.47$ | $36.93 \pm 0.31$ | $35.22 \pm 1.44$ | $38.57 \pm 0.38$ | $37.40 \pm 0.50$ | $36.50 \pm 0.63$ |
| | ProMix | $38.55 \pm 1.10$ | $37.08 \pm 1.54$ | $35.38 \pm 1.74$ | $38.89 \pm 1.49$ | $37.54 \pm 0.84$ | $35.33 \pm 1.84$ |
| | GD$^2$ | $\mathbf{39.80 \pm 0.73}$ | $\mathbf{37.58 \pm 1.10}$ | $\mathbf{35.72 \pm 1.04}$ | $\mathbf{39.97 \pm 0.55}$ | $\mathbf{38.76 \pm 0.74}$ | $\mathbf{38.28 \pm 0.90}$ |

We now bound $\|\boldsymbol{\xi}_i\|_2$. First note that:

$$\mathbb{E}[\boldsymbol{\xi}_i] = \mathbb{E}[\mathbf{x}_i] - \frac{1}{|\mathcal{N}_i|} \sum_{j \in \mathcal{N}_i} \mathbb{E}[\mathbf{x}_j] = \boldsymbol{\mu}_e - \boldsymbol{\mu}_s.$$

We utilize Hoeffding's inequality to bound the deviation of each dimension of $\boldsymbol{\xi}_i$ from its expectation. Let $\mathbf{x}_i[k]$ denote the $k$-th coordinate of $\mathbf{x}$, for $k = 1, \ldots, d$. Then, for each fixed $k$, the set $\{\mathbf{x}_j[k] \mid j \in \mathcal{N}(i)\}$ consists of i.i.d. bounded random variables (by assumption), and by directly applying Hoeffding's inequality, we have:

$$\mathbb{P}\left(|\boldsymbol{\xi}_i[k] - \mathbb{E}[\boldsymbol{\xi}_i[k]]| \geq t_1\right) = \mathbb{P}\left(\left|\mathbf{x}_i[k] - \frac{1}{|\mathcal{N}_i|} \sum_{j \in \mathcal{N}_i} \mathbf{x}_j[k] - \mathbb{E}\left[\mathbf{x}_i[k] - \frac{1}{|\mathcal{N}_i|} \sum_{j \in \mathcal{N}_i} \mathbf{x}_j[k]\right]\right| \geq t_1\right)$$

$$= \mathbb{P}\left(\left|(\mathbf{x}_i[k] - \mathbb{E}[\mathbf{x}_i[k]]) - \left(\frac{1}{|\mathcal{N}_i|} \sum_{j \in \mathcal{N}_i} \mathbf{x}_j[k] - \mathbb{E}\left[\frac{1}{|\mathcal{N}_i|} \sum_{j \in \mathcal{N}_i} \mathbf{x}_j[k]\right]\right)\right| \geq t_1\right)$$

$$\leq 2\exp\left(-\frac{\deg(i) + 1}{2B^2} t_1^2\right)$$

(16)

We now use the following implication: if the $d_2$ norm of the deviation vector satisfies

$$\|\boldsymbol{\xi}_i - \mathbb{E}[\boldsymbol{\xi}_i]\|_2 \geq \sqrt{d}t_1,$$

then at least one coordinate $k \in \{1, \ldots, d\}$ must satisfy

$$|\boldsymbol{\xi}_i[k] - \mathbb{E}[\boldsymbol{\xi}_i[k]]| \geq t_1$$

Therefore, by the union bound over all $d$ dimensions:

$$\begin{aligned}
\mathbb{P}\left(\|\boldsymbol{\xi}_i - \mathbb{E}[\boldsymbol{\xi}_i]\|_2 \geq \sqrt{d}t_1\right) & \\
&\leq \sum_{k=1}^{d} \mathbb{P}\left(|\boldsymbol{\xi}_i[k] - \mathbb{E}[\boldsymbol{\xi}_i[k]]| \geq t_1\right) \\
&\leq 2d \cdot \exp\left(-\frac{\deg(i) + 1}{2B^2}t_1^2\right).
\end{aligned} \tag{17}$$

Now, set $t_1 = \frac{t_2}{\sqrt{d}}$, we obtain the vector norm deviation bound:

$$\mathbb{P}\left(\|\boldsymbol{\xi}_i - \mathbb{E}[\boldsymbol{\xi}_i]\|_2 \geq t_2\right) \leq 2d \cdot \exp\left(-\frac{\deg(i) + 1}{2B^2d}t_2^2\right).$$

By triangle inequality:

$$\|\boldsymbol{\xi}_i\|_2 \leq \|\boldsymbol{\xi}_i - \mathbb{E}[\boldsymbol{\xi}_i]\|_2 + \|\mathbb{E}[\boldsymbol{\xi}_i]\|_2 = \|\boldsymbol{\xi}_i - \mathbb{E}[\boldsymbol{\xi}_i]\|_2 + \|\boldsymbol{\mu}_e - \boldsymbol{\mu}_s\|_2.$$

Hence,

$$\mathbb{P}\left(\|\boldsymbol{\xi}_i\|_2 \geq \frac{t}{\rho(\mathbf{W})}\right) \leq \mathbb{P}\left(\|\boldsymbol{\xi}_i - \mathbb{E}[\boldsymbol{\xi}_i]\|_2 \geq \frac{t}{\rho(\mathbf{W})} - \|\boldsymbol{\mu}_e - \boldsymbol{\mu}_s\|_2\right),$$

as long as $\frac{t}{\rho(\mathbf{W})} - \|\boldsymbol{\mu}_e - \boldsymbol{\mu}_s\|_2 > 0$. Finally, we conclude:

$$\mathbb{P}(\delta_i \geq t) \leq 2d \cdot \exp\left(-\frac{\deg(i) + 1}{2B^2d}\left(\frac{t}{\rho(\mathbf{W})} - \|\boldsymbol{\mu}_e - \boldsymbol{\mu}_s\|_2\right)^2\right).$$

$\square$

## E    Pseudo-code

We describe the overall pipeline of GD$^2$ in Algorithm 1.

## F    Limitation

One limitation of the current framework lies in its focus on static, homogeneous graphs. While the proposed discrepancy-based noise detection mechanism demonstrates strong performance under these settings, its applicability to more complex graph structures remains unexplored. In particular, dynamic graphs introduce temporal evolution in both node features and structure patterns, which may obscure discrepancy signals or introduce new sources of semantic drift. Similarly, heterogeneous graphs involve multiple node and edge types, where semantic alignment across views can be relation-specific and context-dependent. These additional layers of structural and semantic complexity pose significant challenges for robust noise detection. Future work will explore how to adapt the proposed framework to these more general graph scenarios, potentially incorporating temporal modeling techniques or relation-aware discrepancy estimation to maintain effectiveness under such conditions.

---

**Algorithm 1** Pseudo code of GD$^2$

---

**Input**: Node feature matrix $\mathbf{X}$, adjacency matrix $\mathbf{A}$, noisy labels $\tilde{\mathbf{Y}}$, training sets $\mathcal{V}_{\text{train}}$
**Parameter**: Training epochs $E$, clean probability threshold $\tau$, pseudo-label confidence threshold $\gamma$; GNN encoder $f_\theta$, linear classifiers $g_{\phi_1}, g_{\phi_2}$

1: Initialize parameters $\theta$, $\phi_1$, $\phi_2$
2: **for** epoch = 1 to $E$ **do**
3:     *// Noise Detection*
4:     Compute view-specific representations: $\mathbf{Z}_e = f_\theta(\mathbf{X}, \mathbf{I})$, $\mathbf{Z}_s = f_\theta(\mathbf{X}, \mathbf{A})$
5:     Compute predictions: $\mathbf{P}_e = g_{\phi_1}(\mathbf{Z}_e)$,    $\mathbf{P}_s = g_{\phi_2}(\mathbf{Z}_s)$
6:     Measure prediction discrepancy $\delta_i = \|\mathbf{P}_e^{[i]} - \mathbf{P}_s^{[i]}\|_2$,    $i \in \mathcal{V}_{\text{train}}$
7:     Fit GMM on $\{\delta_i\}$ and compute clean probability $\mathbb{P}_{\text{clean}}(s \mid \delta_i)$
8:     Determine clean set $\mathcal{V}_{\text{clean}}$ and noisy set $\mathcal{V}_{\text{noisy}}$ using threshold $\tau$ (Eq. (7))
9:     *// Label Purification*
10:    Compute mixed-view prediction $\mathbf{P}_m = g_{\phi_2}(f_\theta(\mathbf{X}, \tilde{\mathbf{A}}))$
11:    Construct $\mathcal{V}_{\text{purified}}$ and assign pseudo-labels $\hat{\mathbf{Y}}$ (Eq. (9) and Eq. (10))
12:    *// Model Training*
13:    Compute loss terms $\mathcal{L}_e$, $\mathcal{L}_s$, and $\mathcal{L}_m$
14:    Joint optimization of total loss $\mathcal{L} = \mathcal{L}_e + \mathcal{L}_s + \mathcal{L}_m$
15:    Update model parameters $\theta, \phi_1$ and $\phi_2$
16: **end for**

**Output**: Trained encoder $f_\theta$ and classifiers $g_{\phi_1}, g_{\phi_2}$

---

