# OpenReview forum: "GD$^2$: Robust Graph Learning under Label Noise via Dual-View Prediction Discrepancy"
_NeurIPS.cc/2025/Conference — NeurIPS 2025 poster_

### Official Review · Reviewer_aLSG · 2025-06-26

**Clarity:** 3
**Significance:** 2
**Originality:** 3
**Rating:** 4
**Confidence:** 4

**Summary:**

Authors present GD2, which is a GNN training framework for node classification designed to be robust against mislabeled nodes. It uses a Gaussian Mixture Model to identify noisy samples by looking at discrepancies between individual node features and the aggregated features around the nodes. They call disagreements between these two "views" as the dual-view prediction discrepancy. Authors show experimental results across six datasets, and generally show improvements over existing baselines.

**Questions:**

Though the authors run experiments on two heterophilic graphs, I struggle to understand why GD2 would work well here. Wouldn't the model expect different labels between any ego node and its structural neighborhood?

There is quite some sensitivity to tau selection. Optimal value will also depend on the dataset. Is there some adaptive selection technique authors would recommend that might be useful on top of grid search?

How would GD2 perform when there is class imbalance? Presumably the GMM learns noise due to a healthy distribution of labels. Would it incorrectly think minority class samples are noisy?

**Ethical Concerns:**

["NO or VERY MINOR ethics concerns only"]

**Final Justification:**

I was generally impressed with the paper to begin with, and would have liked to see stronger results before advocating for a 5 score. But metrics are mostly marginal, so I'd like to keep the score as a 4.

**Limitations:**

Yes

**Quality:**

3

**Strengths And Weaknesses:**

Strengths: the dual-view mechanism seems intuitive, and grounded in (some) theory. The experiments are well-designed, using different levels of noise to simulate mislabeled nodes, then checking to see how different models react. Some decent ablations as well to help readers understand the importance of individual components in GD2. Overall, the implementation doesn't seem too heavyweight to apply to existing GNNs as well.

Weaknesses: my main reservation with this paper is that discrepancies in labels between the two views could be an outcome of heterophily in the graph. I would like to see more discussion and/or theory behind the specific gains on heterophilic graphs. Perhaps this ties in with the heterogeneous graphs, which the authors have acknowledged as one limitation. Lastly, performance gains are minor, albeit consistent. Usually <~5% relative performance gain.

---

> ### Author Rebuttal · Authors · 2025-07-29
>
> **Response to Q1**
>
> *“My main reservation with this paper is that discrepancies in labels between the two views could be an outcome of heterophily in the graph, rather than label noise... Wouldn't the model expect different labels between any ego node and its structural neighborhood?”*
>
> We appreciate the reviewer’s thoughtful concern. We would like to clarify that our method does not rely on the homophily assumption, and is in fact designed to function under both homophilous and heterophilous graphs.
>
> The core idea of GD2 is to compare the semantic consistency between two information sources:
>
> - The ego-view reflects the class implied by the node’s own features.
> - The structure-view reflects the class pattern inferred from its neighbors via GNN aggregation—not the actual neighbor labels themselves.
>
> Even in heterophilous graphs, nodes of the same class often share similar neighborhood distributions, even if their neighbors belong to different classes. This observation underpins many heterophily-aware GNNs and is supported by prior works [1,2,3]. Therefore, a clean-labeled node should still exhibit alignment between its ego- and structure-view predictions, while noisy nodes often show disagreement across views due to contradictory evidence.
>
> Importantly, GD2 does not penalize cross-view label disagreement per se. Instead, we use the magnitude of the prediction discrepancy as a soft indicator of semantic inconsistency. This formulation allows us to remain effective even when structural labels differ from the ego node’s label—as is typical in heterophilous graphs.
>
> To empirically verify this, we include results on two heterophilous benchmarks: Roman-Empirical and Amazon-Ratings. As shown in Table 1, GD2 achieves consistent improvements over baselines under various noise settings, confirming that our method is robust to heterophily.
>
> [1] Simple and Asymmetric Graph Contrastive Learning without Augmentations, NeurIPS 2023
>
> [2] Is Homophily a Necessity for Graph Neural Networks?, ICLR 2022
>
> [3] Decoupled Self-supervised Learning for Graphs, NeurIPS 2022
>
> **Response to Q2**
>
> *“There is quite some sensitivity to tau selection. Optimal value will also depend on the dataset. Is there some adaptive selection technique authors would recommend that might be useful on top of grid search?”*
>
> Thank you for the insightful question. We would like to clarify that GD2 exhibits stable and reliable performance when $\tau \leq 0.7$, with only minor fluctuations. A clear performance drop is observed only at $\tau = 0.9$, where overly aggressive filtering removes too much supervision. These results indicate that the method is not overly sensitive to $\tau$ as long as it is chosen within a reasonable range.
>
> To go beyond grid search, we recommend selecting $\tau$ using a small clean validation set, by maximizing validation accuracy or minimizing label disagreement. Additionally, this setting is closely related to pseudo-label filtering in semi-supervised learning, where model confidence is used to select reliable training samples. A similar principle can be applied here: the cross-view agreement serves as a proxy for prediction confidence, and $\tau$ can be interpreted as a confidence threshold. This aligns with recent techniques such as SoftMatch [1] and FreeMatch [2], which dynamically select samples based on confidence-aware criteria. Inspired by these works, we believe designing adaptive or learnable thresholding mechanisms is a promising future direction.
>
> [1] SoftMatch: Addressing the Quantity-Quality Tradeoff in Semi-supervised Learning, ICLR’23
>
> [2] FreeMatch: Self-adaptive Thresholding for Semi-supervised Learning, ICLR’23
>
> **Response to Q3**
>
> *“How would GD² perform when there is class imbalance? Presumably the GMM learns noise due to a healthy distribution of labels. Would it incorrectly think minority class samples are noisy?”*
>
> We thank the reviewer for raising this important question. We clarify that GD2 does not assume any balanced label distribution, and its noise detection mechanism is inherently class-agnostic. Specifically, the GMM is fitted over cross-view prediction discrepancy scores, which measure disagreement between ego- and structure-view predictions, rather than over class frequencies or node labels.
>
> This design ensures that the model focuses on semantic inconsistency rather than minority status. A node from a minority class is not more likely to be marked as noisy unless its two views yield inconsistent predictions.
>
> Moreover, we note that the datasets used in our experiments—such as Computers, Photo, and WikiCS—are naturally imbalanced. The table below shows the number of samples per class in three datasets(for brevity, we display only a subset of classes):
>
> | Class | Computer | Photo | WikiCS |
> | --- | --- | --- | --- |
> | 0 | 436 | 369 | 295 |
> | 1 | 2,142 | 1,686 | 667 |
> | 2 | 1,414 | 703 | 2,153 |
> | 3 | 542 | 915 | 1,933 |
> | 4 | 5,158 | 882 | 2,679 |

---

> > ### Comment · Reviewer_aLSG · 2025-08-04
> >
> > Thanks for addressing the comments. While I agree that the improvements appear stable and robust, the relative amounts are still low. I'll keep my recommendation of borderline accept.

---

### Official Review · Reviewer_LmxF · 2025-06-28

**Clarity:** 3
**Significance:** 3
**Originality:** 3
**Rating:** 4
**Confidence:** 3

**Summary:**

This paper introduces GD², a framework for addressing label noise in Graph Neural Networks for node classification. The core idea is to detect noisy labels by leveraging the prediction discrepancy between two distinct views of a node: an "ego-view," constructed from the node's own features, and a "structure-view," derived from the representations of its neighborhood. The framework is further enhanced by a view-specific training strategy, which uses a GMM to identify a set of clean nodes and then generates pseudo-labels for a subset of the noisy nodes. By feeding different supervisory signals (original noisy labels vs. purified pseudo-labels) to the different views, the framework amplifies the discrepancy signal.

**Questions:**

1.	How does the method perform on heterophily graphs where the assumption of neighbor similarity may be violated?
2.	What are the performance limitations of the framework under extremely high noise ratios?
3.	What is the computational complexity and runtime of the GD² framework?

**Ethical Concerns:**

["NO or VERY MINOR ethics concerns only"]

**Final Justification:**

The authors responses have addressed my concerns. Hence I have updated my confidence.

**Limitations:**

The authors discuss the limitation that their work is restricted to static graphs.

**Quality:**

3

**Strengths And Weaknesses:**

Strengths
1.	The paper tackles the problem of label noise, which is a significant challenge in real-world applications of GNNs. Developing methods that improve robustness is of high practical value.
2.	The view-specific training strategy is clever: rather than passively observing the discrepancy between views, the framework proactively amplifies this signal. By feeding original noisy labels to the ego-view and purified labels to the structure-view, the method intentionally creates a larger divergence for mislabeled nodes.
3.	As reported, the proposed GD² framework achieves state-of-the-art performance, outperforming several baselines across multiple datasets and under various noise settings.

Weaknesses
1.	The method's core premise is that a discrepancy between the ego-view and structure-view indicates a noisy label, which is based on the homophily assumption. In Heterophily graphs, where connected nodes are often of different classes, this assumption may be violated. For example, for fraudulent and legitimate users, a correctly labeled node in such a graph could be misidentified as noisy. More discussion on how the framework adapts to Heterophily graphs should be included.
2.	The paper should further discuss the framework's limitations under extremely high noise ratios (e.g., p > 0.5). At such levels, the neighborhood information becomes unreliable, and the GMM may struggle to find a clean separation between the clean and noisy nodes. The experimental results also suggest that the method’s performance advantage over baselines may diminish under high-noise regimes.
3.	The GD² framework is complex. Each epoch involves computing three distinct views, calculating discrepancies for all training nodes, fitting a GMM, and optimizing three separate loss functions. This introduces significant computational and time overhead. The authors should include an analysis of the computational complexity and runtime costs.

---

> ### Author Rebuttal · Authors · 2025-07-29
>
> **Response to W1 & Q1**
>
> *“The method's core premise is that a discrepancy between the ego-view and structure-view indicates a noisy label, which is based on the homophily assumption. In heterophily graphs, where connected nodes are often of different classes, this assumption may be violated…”*
>
> We thank the reviewer for highlighting this important concern. We clarify that our method does not rely on the homophily assumption (i.e., neighbors sharing the same label). Instead, the dual-view design in GD2 distinguishes information sources rather than relying on node similarity:
>
> - The ego-view captures the prediction from a node’s own features.
> - The structure-view aggregates signals from neighbors and reflects the class pattern implied by the neighborhood distribution—not necessarily the classes of neighbors.
>
> This distinction is especially important in heterophilous graphs, where neighbors tend to belong to different classes. Prior works [1,2,3] have shown that nodes from the same class often share similar neighborhood patterns, even under heterophily. Thus, we expect that for a node with a clean label, its ego- and structure-view predictions should still be semantically aligned—i.e., they should agree on the class, despite structural label diversity.
>
> To empirically verify this, we have conducted experiments on two heterophilous graphs (Roman-empirical and Amazon-ratings). As shown in Table 1, GD2 consistently outperforms baselines under label noise, even when neighbor-label similarity is low. This indicates the robustness of our discrepancy-based framework without relying on homophily.
>
> [1] Simple and Asymmetric Graph Contrastive Learning without Augmentations, NeurIPS 2023
>
> [2] Is Homophily a Necessity for Graph Neural Networks?, ICLR 2022
>
> [3] Decoupled Self-supervised Learning for Graphs, NeurIPS 2022
>
> **Response to W2 & Q2**
>
> *“The paper should further discuss the framework's limitations under extremely high noise ratios (e.g., p > 0.5)...”*
>
> Thank you for raising this important point. We agree that when the noise ratio becomes extremely high, label supervision becomes less reliable, and the risk of polluted structural information increases. Nonetheless, we would like to clarify two key aspects of our method that preserve its effectiveness in such settings.
>
> First, our approach is based on a realistic and widely accepted assumption in noisy-label learning: within each true class, the number of correctly labeled nodes is greater than the number of nodes mislabeled into any single incorrect class. That is, although noisy labels exist globally, for any given ground-truth class, its clean samples are still the most concentrated compared to any other individual class. This class-wise imbalance ensures that the structure-view can retain useful semantic signals, even under significant noise.
>
> Second, our method distinguishes clean and noisy labels based on the relative magnitude of cross-view prediction discrepancy, not on the binary presence of discrepancy. Importantly, this separation remains valid even when the neighborhood contains many noisy nodes.
>
> To illustrate this, consider a 3-class classification task under uniform noise with p=0.6. Assume a node whose ground-truth class is 0, and 60% of its neighbors are truly class 0, while the remaining 40% are class 1. After label corruption, the observed neighbor label distribution becomes: 44% class 0, 36% class 1, 20% class 2.
>
> Assuming the ego-view predicts the given label (one-hot), and the structure-view reflects the noisy neighbor label distribution, we have:
>
> - If the node's given label is correct (class 0), then the ego-view is $(1,0,0)$, and the structure-view is $(0.44,0.36,0.20)$, giving a discrepancy of: $||(1, 0, 0) - (0.44, 0.36, 0.20)|| \approx 0.48.$
> - If the label is corrupted to class 1, the discrepancy becomes: $||(0, 1, 0) - (0.44, 0.36, 0.20)|| \approx 0.64.$
> - If corrupted to class 2, the discrepancy is even larger: $||(0, 0, 1) - (0.44, 0.36, 0.20)|| \approx 0.96.$
>
> This example demonstrates that even with p=0.6, noisy-labeled nodes consistently yield higher discrepancy scores, enabling effective separation by GMM.
>
> Finally, while our performance margins over baselines narrow under extreme noise, GD2 still consistently ranks among the top-performing methods, as shown in Table 1. This reflects a fundamental challenge shared by most robust GNNs, rather than a weakness specific to our design.
>
> **Response to W3 & Q3**
>
> *“The GD² framework is complex... The authors should include an analysis of the computational complexity and runtime costs.”*
>
> We thank the reviewer for the suggestion. While GD² introduces dual-view inference, discrepancy modeling, and multi-view optimization, its overall computational cost remains moderate and scales linearly with graph size. We provide a detailed breakdown below, based on Algorithm 1.
>
> 1. **View-specific representation (Line 4):**
>
>     Each epoch involves two GCN forward passes: one with identity matrix $I$ (ego-view), and one with adjacency $A$ (structure-view). For $L$ layers, input feature dimension $F$, and $N=∣V∣$ nodes, the total cost is: $O\big(2L(|E|F + NF^2)\big)$
>
> 2. **Prediction and discrepancy (Lines 5–6):**
>
>     Two linear classifiers produce logits in $O(NFC)$. Then, computing the $\ell_2$ discrepancy for each training node costs $O(NC)$.
>
> 3. **GMM fitting (Line 7):**
>
>     The discrepancy values are 1D scalars; fitting a 2-component GMM via EM costs only $O(N)$.
>
> 4. **Mixed-view inference and pseudo-labeling (Lines 10–11):**
>
>     One additional GCN forward pass adds $O(L(|E|F + NF^2))$. Pseudo-label assignment costs at most $O(NC)$.
>
> 5. **Loss computation and update (Lines 13–15):**
>
>     Three view-specific cross-entropy losses are computed over subsets of nodes, costing $O(NC)$, followed by standard backpropagation with cost similar to one GCN pass.
>
>
> In total, each epoch has complexity: $O\big(3L(|E|F + NF^2) + NFC\big)$.  Compared to a standard GCN with per-epoch cost $O(L(|E|F + NF^2) + NFC)$, GD2 introduces roughly 2–3× overhead due to three view-specific passes and auxiliary steps such as GMM fitting and pseudo-labeling. However, these extra components scale linearly with node count an remain lightweight in practice.
>
> To further demonstrate practical efficiency, we report the total training time (in seconds) of GD2 against representative baselines across datasets in the table below.
>
> | Dataset | CE | ERASE | TSS | GD² |
> | --- | --- | --- | --- | --- |
> | Computer | 54.1 | 215.1 | 92.8 | 202.3 |
> | WikiCS | 30.5 | 152.5 | 55.8 | 160.3 |
>
> We observe that GD2 has comparable runtime to ERASE, another strong noise-robust baseline, and is slower than TSS. However, GD2 significantly outperforms TSS, offering a favorable trade-off between robustness and efficiency.

---

> > ### Comment · Reviewer_LmxF · 2025-08-05
> >
> > Thank you for your responses. It addressed my primary concerns. I will keep my positive score and update my confidence.

---

### Official Review · Reviewer_nnqL · 2025-06-30

**Clarity:** 1
**Significance:** 2
**Originality:** 3
**Rating:** 4
**Confidence:** 3

**Summary:**

In order to mitigate noises in the graph deep learning without i.i.d. node labels, the paper introduces a method to identify the nodes with nosy labels by measuring the distribution of ego-view representation and structure-view representation. The proposed framework then proposes pseudo-labels for the noisy nodes for accurate nose prediction. The paper also validates the method in theory. Finally, the authors evaluated the method on multiple datasets with different distributions and raios of noisy labels.

**Questions:**

1. Please clarify the narrative problems mentioned in the weakness part above.
2. Please further conduct ablation study that only utilize the structure-view as clean representation for node label prediction in both training and inference.

**Ethical Concerns:**

["NO or VERY MINOR ethics concerns only"]

**Final Justification:**

The authors addressed all my concerns.

**Limitations:**

Yes

**Paper Formatting Concerns:**

No Paper Formatting Concerns

**Quality:**

2

**Strengths And Weaknesses:**

Strengths:

1. The paper proposes a novel approach to identify nodes with noisy labels based on the distribution discrepancy between the ego-view and the structure-view.

2. The paper conducted thorough experiments with both various noise ratio and noise distribution.

Weakness:

1. The paper should clarify how the interdependencies among connected nodes in graph affect the existing noise-agnostic works. To the best of my knowledge, NRGNN also took the neighborhood of nodes with similar labels into consideration and connected the labeled and unlabeled nodes for noise-agnostic learning.

2. The notations of the papar should be reorganized. For example, ‘s’ is redefined as clean-label component in line 158 while it was the the footnote for structure-view in the notations above.

3. The main assumption and theoretical justification for now seems not sound for me. First, take the experiment with p = 0.6 as an example. Emperically, for clean-labeld node with 60% noisy neighbors, I believe there would still be distribution discrepancy between the ego-view and the structure-view. In this way, it cannot be taken as an identification of nodes with noisy labels. Second, for the theoretical justification, I blieve the main suppose the authors should prove is that the the distribution discrepancy should exists and only exists in nodes with noisy labels. However in the proving procedure, the result only provides the upper bound of probability of distribution discrepancy, but does provide any direct correlation between the distribution discrepancy the whether the label of the node is noisy or not.

4. Also, as the authors mention that “the structure-view captures topological patterns that remain relatively stable under noise”, why not directly utilize the structure-view as clean representation for node label prediction?

5. The improvement on the results of the provided experiment seem marginal, with only less than 1% accuracy improvement. In this way, the proposed method seems not effective enough for handling the noisy labels of graph nodes.

---

> ### Author Rebuttal · Authors · 2025-07-29
>
> **Response to W1**
>
> *“The paper should clarify how the interdependencies among connected nodes in graph affect the existing noise-agnostic works. To the best of my knowledge, NRGNN also took the neighborhood of nodes with similar labels into consideration and connected the labeled and unlabeled nodes for noise-agnostic learning.”*
>
> We thank the reviewer for this helpful comment. Indeed, several prior works—such as NRGNN and related methods—have utilized local structural signals (e.g., neighborhood similarity, label propagation) to enhance robustness against noisy labels. These methods implicitly leverage interdependencies among connected nodes to smooth predictions or refine graph structure.
>
> However, our work differs in both purpose and formulation. Specifically, most prior approaches use interdependencies as an auxiliary tool to *mitigate* the effects of label noise—e.g., by refining edges, propagating softened labels, or enforcing similarity regularization. In contrast, our goal is to explicitly model the inconsistency between a node's own features and its structural context as a signal for detecting noise.
>
> To this end, we construct two distinct views for each node: (1) the ego-view, reflecting prediction from a node’s own features;(2) the structure-view, reflecting prediction based on aggregated neighbors.
>
> The cross-view discrepancy between them is not used to refine the graph or smooth labels, but to identify which nodes are likely to be mislabeled. This discrepancy serves as a dynamic and model-internal indicator during training, rather than a static preprocessing step.
>
> Therefore, while both our method and prior works acknowledge the importance of interdependencies, our contribution lies in recasting those dependencies as a core signal for noise detection itself, rather than treating them as a denoising aid.
>
> **Response to W2**
>
> *“The notations of the paper should be reorganized. For example, ‘s’ is redefined as clean-label component in line 158 while it was the footnote for structure-view in the notations above.”*
>
> Thank you for pointing this out. We have carefully revised the notation and related equations to ensure that each symbol has a unique and consistent meaning throughout the paper.  All revisions will be reflected in the updated version of the paper.
>
> **Response to W3**
>
> *“The main assumption and theoretical justification for now seems not sound for me… The result only provides the upper bound of probability of distribution discrepancy, but does not provide any direct correlation between the discrepancy and label noise.”*
>
> Thank you for the thoughtful critique. We believe there is a misunderstanding regarding the role of cross-view discrepancy in our framework. We clarify that our method does not rely on the binary presence or absence of discrepancy to determine whether a node is noisy. Instead, our assumption and algorithmic design are based on the distributional difference in discrepancy scores between clean-labeled and noisy-labeled nodes.
>
> Concretely, we compute a cross-view prediction discrepancy—the distance between the ego-view and structure-view predictions. Our key insight is that this discrepancy is statistically higher for noisy-labeled nodes, as their ego-view (aligned with an incorrect label) tends to disagree more with the structure-view (which captures the semantic context via neighbor aggregation). This discrepancy serves as a soft, probabilistic signal, which we then model using a Gaussian Mixture Model (GMM) to separate clean from noisy samples based on their distributional patterns.
>
> To further illustrate this point intuitively, we provide a simple example demonstrating that even under high noise levels, nodes with incorrect labels tend to exhibit higher cross-view discrepancy. Consider a 3-class classification problem under uniform label noise with p = 0.6. Assume a node whose ground-truth class is 0, and 60% of its neighbors are truly class 0, while the remaining 40% are class 1. Due to label corruption, the observed neighbor label distribution becomes:44% class 0, 36% class 1, 20% class 2.
>
> For the sake of simplicity and interpretability, we assume the ego-view prediction is a one-hot vector aligned with the given label, and the structure-view prediction is the distribution of observed neighbor labels.
>
> Under these assumptions:
>
> - If the node's given label is correct (class 0), then the ego-view is $(1,0,0)$, and the structure-view is $(0.44,0.36,0.20)$, giving a discrepancy of: $||(1, 0, 0) - (0.44, 0.36, 0.20)|| \approx 0.48.$
> - If the label is corrupted to class 1, the discrepancy becomes: $||(0, 1, 0) - (0.44, 0.36, 0.20)|| \approx 0.64.$
> - If corrupted to class 2, the discrepancy is even larger: $||(0, 0, 1) - (0.44, 0.36, 0.20)|| \approx 0.96.$
>
> This example demonstrates that nodes with incorrect labels exhibit higher cross-view discrepancy, even when neighborhood noise is significant. The separation between clean and noisy nodes does not depend on the absolute value of discrepancy for any individual node, but rather on the relative distributional differences between clean and noisy groups—which the GMM is designed to exploit.
>
> Regarding the theoretical justification, we appreciate the reviewer’s concern. We clarify that the goal of our theoretical analysis is not to establish a necessary and sufficient condition that links cross-view discrepancy directly and exclusively to label correctness. Instead, the purpose is to demonstrate that, under mild assumptions, clean-labeled nodes tend to have bounded (i.e., low) cross-view discrepancy with high probability—providing a theoretical explanation for the observed distributional separability between clean and noisy samples.
>
> Specifically, Theorem 1 in our paper derives an upper bound on the probability that a clean node exhibits high discrepancy, which formalizes the intuition that clean nodes are less likely to produce inconsistent ego- and structure-view predictions. We do not claim that noisy nodes are the *only* ones with large discrepancy, but rather that noisy nodes are more likely to lie in the high-discrepancy region, making the distributional signal exploitable for separation.
>
> **Response to W4 & Q2**
>
> *“As the authors mention that ‘the structure-view captures topological patterns that remain relatively stable under noise,’ why not directly utilize the structure-view as clean representation for node label prediction? Please further conduct ablation study that only utilize the structure-view as clean representation for node label prediction in both training and inference.”*
>
> Thank you for the insightful question. While the structure-view captures topological patterns that are relatively stable under noise, using it alone would discard valuable node-specific information from the ego-view.
>
> Our final prediction model uses a mixed-view representation that combines both views, but is trained only on the filtered clean subset, where the impact of noisy labels is significantly reduced. In this setting, incorporating the ego-view adds complementary information with minimal noise interference.
>
> The core idea of our method is to use the discrepancy between ego- and structure-view predictions to identify noisy labels. After filtering, combining both views leads to more expressive and robust representations.
>
> Empirically, we show that structure-view alone performs worse than our mixed-view model, validating the benefit of using both.
>
> |  | Computer-Uniform-0.6 | Roman-Uniform-0.6 | Computer-Pair-0.4 | Roman-Pair-0.4 |
> | --- | --- | --- | --- | --- |
> | GD2 | 84.76±0.81 | 51.24±1.68 | 83.16±1.75 | 62.16±1.14 |
> | GD2-structure view only | 82.74±0.60 | 50.17±1.25 | 81.71±1.89 | 60.46±1.87 |
>
> **Response to W5**
>
> *“The improvement on the results of the provided experiment seem marginal, with only less than 1% accuracy improvement.”*
>
> We respectfully disagree and would like to clarify. While some individual improvements appear small in absolute value, GD2 consistently ranks first or second across all datasets and noise settings, demonstrating strong overall robustness and competitiveness.
>
> Notably, under more challenging settings such as asymmetric pair-0.4 noise on Computer, GD2 surpasses ERASE by 3.03%, which is a substantial margin in the context of noisy node classification. Even in cases where the gains are smaller, the improvements are highly consistent and statistically stable across multiple runs, indicating better robustness and generalization.
>
> Overall, GD2 delivers reliable gains with minimal architectural complexity, validating the effectiveness of our discrepancy-based noise filtering approach.

---

> ### Author Response · Authors · 2025-08-06
>
> Dear Reviewer,
>
> We hope this message finds you well. As the discussion phase is entering its final days (less than three days remaining), we noticed that we have not yet received your follow-up comments regarding our paper.
>
> We would sincerely appreciate it if you could let us know whether our rebuttal has addressed your concerns, or if there are any remaining issues we could further clarify. Your feedback would be greatly appreciated and would help us improve our work.
>
> Thank you again for your time and thoughtful reviewing.

---

> > ### Comment · Reviewer_nnqL · 2025-08-07
> >
> > Thanks for your detailed responses. But I still have some concerns.
> >
> > **W1** I understand your intention to differentiate from prior work by investigating label inconsistency between node features and structural contexts. Please elaborate on practical scenarios where such feature-structure label inconsistency occurs, with concrete application examples.
> >
> > **W3** First, I would like to point out that if we adopt your assumption at w1 regarding the existence of label inconsistency between node features and their structural context, the structural view-based labels obtained through aggregation mechanisms might inherently produce inconsistencies even when the original labels are correct. This phenomenon could lead to erroneous identification of noisy labels through the aggregation process, which was the core concern for what I raised in w1. Secondly, since "noisy nodes are not the only ones with large discrepancy" as you mentioned, wouldn't this discrepancy-based identification method for noisy nodes inevitably introduce substantial false positives?

---

> > > ### Author Response · Authors · 2025-08-07
> > >
> > > Thank you for your continued engagement and thoughtful questions.
> > >
> > > We would like to further clarify the intuition and key insight behind our method. Our approach is **not** based on whether prediction inconsistency between node features and their structural context exists, **nor** on the absolute magnitude of inconsistency for any individual node, regardless of how it compares to others. **Instead, we emphasize its distributional pattern across all nodes in the graph: when a node is mislabeled, the inconsistency between its ego-view and structure-view predictions tends to be *statistically and consistently larger* than that of correctly labeled nodes.** This distinction is **not** deterministic per node but **emerges across the population**, which our method explicitly exploits to distinguish clean and noisy samples. Our goal is not to make exact decisions for individual nodes, but to utilize reliable statistical signals for effective separation overall.
> > >
> > > ### **W1**
> > >
> > > Feature-structure inconsistency frequently arises in real-world graphs due to label noise from human error, weak supervision, or incomplete metadata. Below are representative scenarios:
> > >
> > > - **Social networks:** A user may be mislabeled with an incorrect interest group due to noisy or misleading profile information (e.g., sparse activity, ambiguous interests). While the ego-view prediction reflects these corrupted features, the structure-view—based on connections to a consistent community—predicts a different label, creating a strong cross-view discrepancy.
> > > - **Product networks:** A product may be mislabeled because of incomplete or erroneous textual/visual descriptions (e.g., poor metadata or misleading title). The ego-view aligns with these noisy features, but the structure-view—based on co-purchased or taxonomically related items—suggests the correct category, exposing the label inconsistency.
> > > - **Biomedical networks:** A protein or compound may be mislabeled as one function (e.g., metabolic) due to atypical or noisy molecular features (e.g., rare motifs or incomplete annotations). Its ego-view prediction supports this incorrect label, whereas its neighbors participate in a different biological process (e.g., signaling), leading to a conflicting prediction from the structure-view.
> > > - **Political opinion networks:** In networks where users with opposing political views interact (e.g., via debates or retweets), a user may be labeled incorrectly based on a small number of ambiguous or ironic posts. The ego-view, relying solely on this noisy content, supports the wrong label, while their interaction graph (e.g., frequent engagement with users of the opposing view) leads the structure-view to predict a different stance.
> > >
> > > ### **W3**
> > >
> > > While aggregation mechanisms might introduce inconsistencies even for clean nodes, this does not undermine our method. We do **not** treat the presence of inconsistency as direct evidence of label noise. Instead, our approach relies on the overall distribution of discrepancy magnitudes—noisy nodes are more likely to fall in the high-discrepancy region, which the GMM leverages for statistical separation. This distributional modeling avoids misidentifying clean nodes due to isolated fluctuations.
> > >
> > > As for false positives: our method does not rely on whether a node’s discrepancy is large in isolation, but on the **distributional difference** between clean and noisy nodes. While clean nodes may exhibit large discrepancy, **noisy nodes tend to have even larger discrepancy values overall**, creating a clear statistical margin (see Figure 1) that makes substantial false positives unlikely. This separation is further reinforced by technical strategies such as high-confidence filtering, asymmetric supervision, and selective pseudo-labeling, which collectively reduce potential misclassifications.

---

> > > > ### Comment · Reviewer_nnqL · 2025-08-08
> > > >
> > > > Thank you for addressing all my concerns. I will adjust my score to borderline accept.

---

### Official Review · Reviewer_Kyqg · 2025-07-02

**Clarity:** 3
**Significance:** 3
**Originality:** 3
**Rating:** 4
**Confidence:** 3

**Summary:**

The paper proposes GD2, a robust graph learning framework for noisy data. The model utilizes two distinct graph views, the ego-view and the neighbor-view, and their discrepancy, to filter out wrongly labeled data and relabel them during training to achieve robust training.

**Questions:**

See weaknesses.

**Ethical Concerns:**

["NO or VERY MINOR ethics concerns only"]

**Final Justification:**

The paper proposes GD2, a robust graph learning framework that uses dual graph views and their prediction discrepancy to filter noisy labels and progressively relabel them during training. The idea is novel, theoretically grounded, and empirically effective. The method is clearly described and supported by strong results on several benchmark datasets. However, there are a few drawbacks: first, while the rebuttal clarifies the bootstrapped nature of clean/noisy separation, the method still relies on potentially unstable early predictions. Additionally, although the evaluation is broader after the rebuttal, it remains limited in the diversity of graph types. The threshold parameter τ is shown to be moderately robust, but its selection still requires a clean validation set or heuristic tuning; an adaptive or automated mechanism would improve usability.

**Limitations:**

yes

**Quality:**

3

**Strengths And Weaknesses:**

Strengths:

1. The method is novel and is theoretically guaranteed.
2. The empirical result is strong and gives good performance.
3. The paper is clear in explaining the method.

Weaknesses:

1. It is unclear how the model can reliably distinguish clean vs noisy data at the beginning of training, when both ego- and neighbor-view predictions may be unstable.

2. The experiments are conducted on a small number of benchmarks, lacking large-scale datasets [1].

3. The method is only validated with GCNs; it is unknown whether the approach remains effective when applied to other widely used GNN architectures like GAT, GraphSAGE, or transformer-based models.

4. The clean node threshold (τ) varies across datasets, but the paper does not offer a principled or simple strategy for choosing it automatically.

[1] Lim, Derek, et al. "Large scale learning on non-homophilous graphs: New benchmarks and strong simple methods." *Advances in neural information processing systems* 34 (2021): 20887-20902. datasets are too few

---

> ### Author Rebuttal · Authors · 2025-07-29
>
> **Response to W1**
>
> *“It is unclear how the model can reliably distinguish clean vs noisy data at the beginning of training, when both ego- and neighbor-view predictions may be unstable.”*
>
> We appreciate the reviewer’s concern. Our method is designed in a bootstrapped and iterative manner rather than relying on accurate clean/noisy separation at the initial stage. Specifically, as shown in the Pseudo-code in the Appendix E, GD2 updates the clean node set at every training epoch based on the discrepancy between dual-view predictions. This process allows the model to gradually refine its belief about clean samples as training progresses, leveraging the co-training-style evolution of ego-view and neighbor-view representations. Such a progressive filtering mechanism is common and effective in robust learning frameworks under uncertain supervision.
>
> **Response to W2**
>
> *“The experiments are conducted on a small number of benchmarks, lacking large-scale datasets.”*
>
> Thank you for the suggestion. We have included additional experiments on the large-scale ogbn-arxiv dataset. Our method achieves consistent gains under both symmetric and asymmetric label noise settings, showing its scalability and effectiveness beyond the original benchmark datasets.
>
> |  | Uniform-0.2 | Uniform-0.4 | Pair-0.2 | Pair-0.4 |
> | --- | --- | --- | --- | --- |
> | GCN | 60.16±0.57 | 54.12±1.12 | 62.18±0.13 | 50.81±1.74 |
> | ERASE | 62.97±1.04 | 57.60±1.76 | 65.48±0.92 | 52.06±1.96 |
> | GD2 | 65.82±1.16 | 59.75±1.52 | 67.51±1.07 | 54.88±1.69 |
>
> **Response to W3**
>
> *“The method is only validated with GCNs; it is unknown whether the approach remains effective when applied to other widely used GNN architectures like GAT, GraphSAGE, or transformer-based models.”*
>
> Thank you for pointing this out. We would like to clarify that GD2 is inherently architecture-agnostic and does not rely on the specifics of any GNN variant. In fact, message-passing GNNs—including GCN, GAT, GraphSAGE, and Transformer-based models—can be abstracted as consisting of two core steps:
>
> (1) Aggregation: gathering information from a node’s local neighbors or global context;
>
> (2) Update: combining the aggregated message with the node’s current embedding.
>
> Our dual-view construction leverages this decomposition:
>
> - The ego-view is computed by skipping the aggregation step, relying only on the node’s own features and performing the update step independently.
> - The structure-view is derived by removing the ego-node’s self-loop during aggregation (i.e., excluding its own embedding from the neighborhood), thus obtaining a purely structural context from neighbors.
>
> In this way, GD2 defines two complementary perspectives via selective aggregation operations, which are supported in virtually all message-passing GNNs.
>
> To demonstrate generalizability, we have included new experiments on Computer using GAT and SGFormer [1] as the backbone architecture (see Table below). GD2 consistently improves robustness on these architectures, confirming that our method is not confined to GCNs.
>
> |  | Uniform-0.2 | Uniform-0.4 | Pair-0.2 | Pair-0.4 |
> | --- | --- | --- | --- | --- |
> | GCN | 85.33±0.45 | 82.68±0.66 | 84.11±0.51 | 77.85±1.89 |
> | GD2-GCN | 86.91±0.56 | 86.26±0.71 | 86.88±0.55 | 83.16±1.75 |
> | GAT | 85.08±0.22 | 83.03±0.36 | 84.50±0.81 | 78.51±1.34 |
> | GD2-GAT | 86.69±0.68 | 85.94±0.40 | 86.95±0.82 | 83.91±1.50 |
> | SGFormer | 84.95±0.47 | 81.52±0.56 | 83.20±0.82 | 78.13±1.67 |
> | GD2-SGFormer | 87.21±0.93 | 85.51±0.86 | 87.04±0.19 | 82.83±1.10 |
>
> [1] SGFormer: Simplifying and Empowering Transformers for Large-Graph Representations. NIPS’23
>
> **Response to W4**
>
> *“The clean node threshold ($\tau$) varies across datasets, but the paper does not offer a principled or simple strategy for choosing it automatically.”*
>
> Thank you for the valuable suggestion. The threshold $\tau$ determines how strictly GD2 filters out noisy nodes based on the agreement between ego-view and structure-view predictions.
>
> As shown in Figure 3, GD2 achieves stable and reliable performance when $\tau≤0.7$, with only minor fluctuations. A noticeable performance drop occurs at $\tau = 0.9$, where overly conservative filtering removes too much supervision. This indicates that the method is not overly sensitive to $\tau$, as long as it is set within a moderate range.
>
> In practice, $\tau$ can be selected using a small clean validation set, by maximizing validation accuracy or minimizing disagreement with clean labels. Alternatively, this setting is closely related to pseudo-label filtering in semi-supervised learning, where model confidence is used to select high-quality training signals. A similar principle can be applied here: cross-view agreement serves as a proxy for prediction confidence, and $\tau$ can be set to retain nodes with high agreement. This idea aligns with strategies used in recent works such as SoftMatch [1] and FreeMatch [2]. Inspired by these works, we believe designing adaptive or learnable thresholding mechanisms is a promising future direction.
>
> [1] SoftMatch: Addressing the Quantity-Quality Tradeoff in Semi-supervised Learning, ICLR’23
>
> [2] FreeMatch: Self-adaptive Thresholding for Semi-supervised Learning, ICLR’23

---

> > ### Comment · Reviewer_Kyqg · 2025-08-03
> >
> > I thank the author for the detailed replies. My concerns are resolved and I will keep my positive score of 4.

---

### Author Response · Authors · 2025-08-08
**General Response**

**We sincerely thank all reviewers for the time, effort, and thoughtful feedback provided during the review process.**

We greatly appreciate your engagement and are glad that several aspects of our work were positively recognized, including:

- Novel and clever design: Reviewers appreciated the originality of the dual-view discrepancy framework, and found the view-specific training strategy to be an intuitive and effective way to amplify the noise signal.
- Theoretical support: our method is backed by theoretical analysis showing that clean nodes exhibit bounded prediction discrepancy, providing a principled foundation for our approach.
- Strong and thorough experiments: our method achieves state-of-the-art performance across multiple datasets and noise settings.

During the rebuttal period, we carefully addressed all reviewer concerns and validated our claims with additional experiments. Our key responses are summarized as follows:

- We clarified that our method adopts a bootstrapped training scheme and demonstrated its scalability and generality through additional experiments.
- We explained that our method fundamentally differs from prior works by explicitly modeling feature-structure inconsistency. We further elaborated on the key insight and theoretical justification behind our approach, supported by concrete examples. Additional ablations show the effectiveness of the mixed-view design.
- We clarified that our method does not rely on the homophily assumption and remains effective under high noise. We also discussed the computational complexity and runtime of GD2, which introduces only moderate overhead.
- We showed that GD2 is insensitive to the hyperparameter $\tau$ within a reasonable range and discussed potential strategies for adaptive tuning. Finally, we clarified that GD2 does not assume any balanced label distribution, and its noise detection mechanism is inherently class-agnostic.

Once again, we are deeply grateful for the reviewers' insights and suggestions.

---

### Note · Authors · 2025-08-12

**We sincerely thank the PCs, SACs, ACs, and all reviewers for their time, effort, and thoughtful feedback throughout the review process.**

We propose GD2, a robust graph learning framework that detects label noise via dual-view prediction discrepancy and amplifies it through view-specific training, with theoretical guarantees and strong empirical results. The findings show that modeling feature and structural dependencies enables effective label noise detection and highlights the potential of graph-specific relational patterns to enhance robustness.

We are encouraged that several contributions were positively recognized by the reviewers, including:
- **Novel and clever design**: Reviewers appreciated the originality of the dual-view discrepancy framework, and found the view-specific training strategy to be an intuitive and effective way to amplify the noise signal.
- **Theoretical support**: GD2 is backed by theoretical analysis showing that clean nodes exhibit bounded prediction discrepancy, providing a principled foundation for our method.
- **Strong and thorough experiments**: GD2 achieves state-of-the-art performance across multiple datasets and noise settings.

During rebuttal, we addressed all reviewer concerns and validated our claims with additional experiments. Key responses are summarized as follows:
- We clarified that GD2 adopts a bootstrapped training scheme and demonstrated its scalability and generality through additional experiments.
- We explained that GD2 fundamentally differs from prior works by explicitly modeling feature–structure inconsistency. We further elaborated on the key insight and theoretical justification behind our approach, supported by concrete examples. Additional ablations show the effectiveness of the mixed-view design.
- We clarified that GD2 does not rely on the homophily assumption and remains effective under high noise. We also discussed the computational complexity and runtime, which introduces only moderate overhead.
- We showed that GD2 is insensitive to the hyperparameter within a reasonable range and discussed potential strategies for adaptive tuning. Finally, we clarified that GD2 does not assume any balanced label distribution, and its noise detection mechanism is inherently class-agnostic.

Once again, we thank the reviewers for their constructive feedback, which has helped us further refine our work. We sincerely hope that our efforts and the contributions of GD2, will earn your support for acceptance.

---

### Decision · Program_Chairs · 2025-09-17

**Decision:**

Accept (poster)

**Comment:**

This paper introduces a robust graph learning framework for noisy data, which measures the distribution of ego-view representation and structure-view representation. The reviewers pointed out the novelty of the proposed method and thorough experiments with both various noise ratios and noise distributions. The paper is also well-written with strong theoretical justification.

The reviewers also pointed out some concerns, including a lack of clarification on how the interdependencies among connected nodes in the graph affect the existing noise-agnostic works, and the core premise. The authors provide extensive discussion to clarify these concerns and all the reviewers gave positive scores. I lean towards the acceptance of the papers and hope the rebuttal content can be well included in the camera-ready version.